# Has working-age morbidity been declining? Changes over time in survey measures of general health, chronic diseases, symptoms and biomarkers in England 1994–2014

Ben Baumberg Geiger 

School of Social Policy, Sociology and Social Research (SSPSSR), University of Kent, Canterbury, UK

**Correspondence to**
Dr Ben Baumberg Geiger;
b.b.geiger@kent.ac.uk

## ABSTRACT

**Objectives** As life expectancy has increased in high-income countries, there has been a global debate about whether additional years of life are free from ill-health/disability. However, little attention has been given to changes over time in morbidity in the *working-age* population, particularly outside the USA, despite its importance for health monitoring and social policy. This study therefore asks: what are the changes over time in working-age morbidity in England over two decades?

**Design, setting and participants** We use a high-quality annual cross-sectional survey, the Health Survey for England (HSE) 1994–2014. HSE uses a random sample of the English household population, with a combined sample size of over 140 000 people. We produce a newly harmonised version of HSE that maximises comparability over time, including new non-response weights. While HSE is used for monitoring population health, it has hitherto not used for investigating morbidity as a whole.

**Outcome measures** We analyse all 39 measures that are fully comparable over time—including chronic disease diagnoses, symptomatology and a number of biomarkers—adjusting for gender and age.

**Results** We find a mixed picture: we see improving cardiovascular and respiratory health, but deteriorations in obesity, diabetes, some biomarkers and feelings of extreme anxiety/depression, alongside stability in moderate mental ill-health and musculoskeletal-related health. In several domains we also see stable or rising chronic disease *diagnoses* even where *symptomatology* has declined. While data limitations make it challenging to combine these measures into a single morbidity index, there is little systematic trend for declining morbidity to be seen in the measures that predict self-reported health most strongly.

**Conclusions** Despite considerable falls in working-age mortality—and the assumptions of many policy-makers that morbidity will follow mortality – there is no systematic improvement in overall working-age morbidity in England from 1994 to 2014.

## INTRODUCTION

As life expectancy has increased in high-income countries, there has been a global debate about whether additional years of life are free from

### Strengths and limitations of this study

► We provide a robust analysis of changes over time in morbidity in England for 39 measures across two decades using the Health Survey for England (HSE).
► We include every morbidity measure for which consistent comparisons over time can be constructed in the HSE.
► We take care to maximise comparability over time, including constructing new non-response weights.
► However, response rates for each stage of the HSE have declined over time, and it is impossible to rule out changing non-response biases.
► There are also several dimensions of morbidity for which there is little trend data in HSE.

ill-health/disability. It is now largely accepted that old-age disability has declined in the USA (although varying by age/method),[1 2] although chronic illness increased,[3] and the picture beyond the USA is more mixed.[4–6] Yet, this research agenda has not been matched by similar attention to changes over time in morbidity in the *working-age* population. In the absence of direct evidence, policy-makers have often made claims based on self-reports of general health[6–8] which we know are unreliable.[9 10] The lack of evidence is even more problematic within social security, where many policy-makers have assumed that working-age morbidity *must* have improved in recent decades given improvements in mortality (despite the potential for declining mortality to coexist with rising morbidity)[6]—and that therefore high/rising levels of claims are not 'genuine'.[11 12]

Almost the only direct evidence on changes over time in working-age morbidity in high-income countries comes from the USA. Contrary to policy-maker expectations, these studies have generally found *deteriorating*

**BMJ**

morbidity since the mid-1990s, particularly activities of daily living and physical functioning.[13–16] Other studies have focused on the older working-age population with similar results.[2 17] Again, not all measures show deteriorations, and not all studies come to identical conclusions,[18] but there is little sign of any improvement in morbidity among working-age Americans—despite a 23% fall in working-age mortality 1993–2013 (online supplementary appendix 1). Outside of the USA, there is a paucity of evidence, but from the limited evidence that exists, there is again little sign of improving morbidity.[19–22]

This study therefore asks: is there empirical support for the hypothesis that working-age morbidity in England has declined? (H$_1$). Or does the evidence support alternative hypotheses of stable (H2) or even declining (H3) morbidity? We answer this using the Health Survey for England (HSE), a high-quality Government survey with a combined sample of 140 000 individuals. We examine 39 specific aspects of morbidity rather than reducing morbidity to a single measure, partly because these produce more reliable trends, and partly to capture the multidimensional nature of morbidity.[23] However, we conclude by examining the broad picture of morbidity change, and how far this supports the competing hypotheses.

This analysis makes two contributions. First, we provide one of the few systematic analyses of changes over time in working-age morbidity in any high-income country outside the USA. Second, we supplement self-report measures with 10 'biomarkers' which are particularly valuable for showing genuine changes over time (rather than merely changes in how people describe their health), but which have rarely been examined alongside self-reported working-age morbidity trends (Martin *et al*[24] being an exception).

## Data and methods

This section follows the Strengthening the Reporting of Observational Studies in Epidemiology cross sectional reporting guidelines.[25]

### Data source

Robust evidence of change over time requires consistently collected, high-quality data. We use the HSE, an annual government-sponsored cross-sectional survey of 3000–11 000 adults with no proxy responses.[26–47] A particular advantage is that the interview is followed by a nurse visit which in selected years also includes a blood sample. Nevertheless, there are challenges in analysing change in HSE:

► First, HSE was run by the Government Office of Population Censuses and Surveys in 1991–93, before changing to NatCen in 1994. We focus on 1994–2014 given evidence of a discontinuity at this point.

► Second, topic coverage of HSE varies year-to-year, accompanied by changes in question wording/ filtering. Based on a systematic search of HSE questions, we have included every morbidity measure that is comparable over a significant duration. Even for measures that have been previously been analysed

(eg, body mass index),[48] this new analysis uncovered further discontinuities (online supplementary appendices 2 and 3).

► Third, HSE excludes those in communal establishments. While a smaller problem for the working-age population than older ages,[2] we minimise the impact of rising university attendance by focusing on those aged 25+ (online supplementary appendix 3). The upper limit of the working-age population is set to 59 (women) and 64 (men) to match state pension ages at the start of the period.

► Fourth, HSE supplies non-response weights from 2003. However, there had been a substantial decline in response rates prior to the introduction of weights, particularly for blood samples (from 53.3% 1994 to 39.9% 2003; online supplementary appendix 3). We therefore reduce non-response biases by creating new non-response weights, described in online supplementary appendix 3.

The resulting sample sizes for the various stages of data collection are shown in online supplementary appendix 3. Our dataset substantially extends an existing HSE time-series dataset (UK Data Archive SN7025); the code enabling other researchers to assemble this extended time-series dataset are freely available.[49]

### Patient involvement

As this is a health monitoring study using secondary data, patients were not directly involved. However, from previous discussions we are aware that the study will be of interest to patient/disability advocacy groups, who will receive jargon-free summaries of the research.

### Measures

We cannot interpret changes over time correctly without understanding different ways of operationalising 'morbidity'.[1] General health/disability measures—for example, '*How is your health in general?*'—are a simple way of measuring morbidity with a single indicator, and clearly do capture something meaningful.[50] However, their generality means that despite consistent question wording, different people may interpret questions or response options differently (eg, what 'good' health refers to).[51 p218–24] This can even occur *within* individuals, if they change their internal standards of measurement over time (contributing to 'response shift').[52] Numerous causal factors contribute to variable comprehension/ reporting, ranging from the experience of ill-health itself[52] to non-health factors such as social security incentives,[53] gendered-related and age-related expectations, and medicalisation.[54]

These inconsistencies mean that general health/ disability measures are inadequate for answering our question: trends in such measures can differ wildly between different surveys covering nominally the same concept and population, for example, for disability in England[9] or self-rated health in the USA.[10] Indeed, the HSE itself shows that England has experienced deteriorating 'bad

general health' at the same time as activity limitations have fallen (changes over time in seven general HSE health/disability measures are available in online supplementary appendix 4). Moreover, single indicator measures are potentially misleading in that they gloss over the multidimensional nature of morbidity.[1]

To robustly answer our research question, we must instead focus on more *specific* morbidity measures that capture multiple aspects of morbidity. Our systematic search found 39 such measures that are comparable over time: these are summarised in table 1, with further details in online supplementary appendix 5. (A further 29 measures are also included in online supplementary appendix 6; this includes eight sub-components of measures in the main text, 16 reports of ever having a condition even if this not recent, and five other categories of longstanding illness (LSI).) These specific morbidity measures can be grouped into three types which have different strengths and weaknesses with respect to our question:

1. *Medical labels:* some measures are based on medical labels, either diagnosed chronic diseases or self-reported types of LSI. (Those reporting a LSI were asked, *'what is the matter with you?'*; up to six responses were then coded by the interviewer based on the International Classification of Diseases (ICD)). These are imperfect measures of morbidity[55] as they partly reflect healthcare systems and medicalisation more broadly, both of which change over time. Nevertheless, they are an important element of morbidity as they have real consequences via increasing awareness/labelling of people's experiences.
2. *Symptom-based:* some measures are based on self-reports of ill-health symptoms or specific domains of activity limitations. These measures are either single items (eg, pain, anxiety/depression) or validated symptom scales (eg, the Rose angina scale,[56 57] General Health Questionnaire (GHQ) psychiatric distress).[58] The more specific and concrete nature of these measures prima facie makes them more likely to be interpreted consistently over time than medical labels and general measures. Others have reached a similar conclusion for comparisons across place,[55] particularly for disability measurement,[59 60] where the Washington Group on Disability Statistics—a UN agency founded in 2001—have brokered a consensus that cross-country disability comparisons should be based on multiple measures of specific activity limitations.[61 62] We should nevertheless note that there is no guarantee that a given symptom/impairment-based question will be interpreted identically over time.[63 64]
3. *Biomarkers*—that is, objective measures of biological or physiological measures—have considerable strengths in analysing change, as they largely avoiding reporting biases that are likely to vary between socioeconomic groups and over time.[65] They do this at the price of an indirect and sometimes still-debated relationship to morbidity (see online supplementary appendix 5), and do not cover several important morbidity domains (eg, we lack good biomarkers for mental distress, pain and fatigue).

These three types of measures are therefore complementary in understanding changing morbidity: biomarkers are least likely to be affected by changing respondent interpretations over time, but do not capture morbidity well; symptom-based measures capture morbidity well and are reasonably (if still imperfectly) reliable; and label-based measures are flawed in capturing symptoms/limitations but do enable us to capture whether people consider themselves to have a medical condition.

## Analysis

In the first instance, we look at unadjusted changes over time in each morbidity indicator, showing the actual levels of morbidity found in the population. However, we primarily focus on changes after adjustment for sex and age (following others),[66 67] akin to standardising for the age-sex composition of the population. Given that our aim is to *describe* changes rather than to explain them, we do not further adjust for potential causal influences on morbidity that are likely to vary over the period, such as employment over economic cycles. This is a task for future research, but we should note that such analysis is possible using our publicly-available time-series dataset that includes *inter alia* employment status, education and region.

We chose to examine discrete changes from the start to the end of available data for each measure, rather than using linear or non-linear trend terms. Given our aims of informing policy debates, this has three advantages: a discrete change is simple to interpret; it is compatible with the different start/end years available for different measures; and it does not require any assumptions about the functional form of trends (linear trends are particularly unlikely given the role of non-linear economic cycles). Individual survey years are grouped into 3–4-year periods to increase sample size and precision, but single-year prevalence is given in online supplementary appendix 7. Given our binary outcome measures, we use logistic regression models with the following form:

$$y_i = \text{logit}[\beta_1 \text{period}_i + \beta_2 \text{age}_i + \beta_3 \text{male}_i + \beta(\text{age}_i * \text{male}_i)]$$

…where **period**$_i$ refers to a vector of period dummy variables (covering all periods in which there were any observations: 1994–1996, 1997–2000, 2001–2003, 2004–2007, 2008–2010 and 2011–2014); $\beta_1$ is a vector of our primary outcome coefficients showing change between each period and the earliest available period; **age**$_i$ refers to a vector of age dummy variables; male$_i$ refers to a binary gender dummy variable and $\beta_2$, $\beta_3$ and $\beta_4$ refer to the coefficients on age, gender and their interaction, respectively. We present average marginal effects rather than odds ratios, partly because these are simple to understand—odds ratios have no easy real-world interpretation for policy-makers—but primarily because odds ratios are not fully comparable across different models, and cannot therefore underpin our comparison of changes over time between indicators.[68]

**Table 1** HSE morbidity measures

| Category | Measure | Type* | Operationalisation (years available) |
|---|---|---|---|
| CVD | High BP LSI† | L | Hypertension reported as LSI (1994–2011) |
| | Recent high BP | L | Still has (or on medication for) doctor-diagnosed hypertension (1994–2013) |
| | Biomarker high BP | B | Systolic BP ≥140 mm Hg and diastolic BP ≥90 mm Hg (1994–2013) |
| | High total cholesterol | B | Total cholesterol ≥5 mmol/L (1994–2012) |
| | Low HDL cholesterol | B | HDL cholesterol ≤1 mmol/L (1998–2013) |
| | Recent heart attack /stroke | L | Doctor-diagnosed heart attack or stroke in past 12 months (1994–2011) |
| | Recent angina | L | Doctor-diagnosed angina in past 12 months (1994–2011) |
| | Ischaemic heart/stroke LSI† | L | Stroke, heart attack or angina reported as LSI (1994–2011) |
| | Heart attack symptoms | S | Ever had severe pain across chest for ½ hour (1994–2011) |
| | Mini stroke (TIA) symptoms | S | Attack of weakness/slurred speech/blurred vision in past 12 months (2003–2011) |
| | Angina symptoms | S | Rose Angina scale definition of angina symptoms (1994–2011) |
| | Any recent CVD | L | Doctor-diagnosed heart condition (exc. hypertension) in past 12 months (1994–2011) |
| | Any CVD LSI† | L | Any CVD reported as LSI (1994–2011) |
| Respiratory | COPD symptoms | S | Regular cough and phlegm for at least 3 months each year (1995–2010) |
| | Lifetime diagnosed asthma | L | Ever had doctor-diagnosed asthma (1995–2010) |
| | Asthma LSI† | L | Asthma reported as LSI (1994–2011) |
| | Breathlessness-grade 2 | S | Short of breath when hurrying up walking uphill (1995–2010) |
| | Breathlessness-grade 3 | S | Short of breath when walking on level ground (1995–2010) |
| | Recent wheezing/asthma | S | Wheezing, whistling in chest or asthma attack in past 12 months (1995–2010) |
| | Wheezing stopping sleep | S | Woken 1+times/week by wheezing/whistling in chest in last 12 months (1994–2010) |
| Obesity and diabetes | BMI-underweight | B | BMI ≤18.5 kg/m² (1994–2013) |
| | BMI-obese | B | BMI ≥ 30 kg/m² (1994–2013) |
| | High waist-hip ratio | B | Waist-hip ratio of >1 for men and >0.85 for women (1994–2013) |
| | Recent diabetes | L | Currently taking medication for doctor-diagnosed diabetes (1994–2013) |
| | Diabetes LSI† | L | Diabetes reported as LSI (1994–2011) |
| | High-glycated haemoglobin | B | $HbA_{1C}$ ≥48 mmol/mol (2003–2013) |
| Mental health | Mental health LSI† | L | Mental health reported as LSI (1994–2011) |
| | Psychiatric distress (GHQ) | S | 4+ negative symptoms from 12-item GHQ (1994–2014) |
| | Anxiety/depression-moderately | S | At least moderately anxious/depressed today (1996–2014) |
| | Anxiety/depression-extremely | S | Extremely anxious/depressed today (1996–2014) |

Continued

**Table 1** Continued

| Category | Measure | Type* | Operationalisation (years available) |
|---|---|---|---|
| Activity limitations and musculo skeletal | Problems walking today | S | Has at least some problems walking about today (1996–2014) |
| | Locomotor limitation | S | Can't walk far/bend down/go up or down stairs without resting (1996–2001) |
| | Problems washing/dressing today | S | Has at least some problems washing/dressing today (1996–2014) |
| | Self-care limitation | S | Difficulty with one of six everyday activities (eg, feeding, dressing) (1995–2001) |
| | Pain-any | S | Has at least some pain or discomfort today (1996–2014) |
| | Pain-extreme | S | Has extreme pain or discomfort today (1996–2014) |
| | Arthritis LSI† | L | Arthritis or rheumatism reported as LSI (1994–2011) |
| | Other musculoskeletal LSI† | L | Other musculoskeletal condition reported as LSI (1994–2011) |
| Sensory and communication | LSI eye or ear | L | Eye or ear condition reported as LSI (1994–2011) |
| | Hearing limitation | S | Cannot follow TV programme at volume others find acceptable (1995–2001) |
| | Seeing limitation | S | Cannot see well enough to recognise friend across the road (1995–2001) |
| | Communicating limitation | S | Have problem communicating with other people (1995–2001) |
| Other biomarkers | Raised CRP | B | CRP >3 mg/L (1998–2009) |
| | Raised fibrinogen | B | Fibrinogen >4 mg/L (1998–2009) |
| | Anaemia | B | Haemoglobin <13 g/dL for men and <12 g/dL for women (1994–2009) |
| | Iron deficiency | B | Serum ferritin <45 ng/mL (1994–2009) |

See online supplementary appendix 5 for full details on all measures.

*Measure type key: L=medical label; S=symptom-based; B=biomarker.

†Particular causes of LSI come from the open question, 'what is the matter with you?' Up to 6 responses are then coded by the interviewer into a coding frame based on ICD.

BMI, body mass index; COPD, chronic obstructive pulmonary disease; CRP, C-reactive protein; CVD, cardiovascular disease; GHQ, General Health Questionnaire; HDL, high density lipoprotein; LSI, longstanding illness; TIA, transient ischaemic attack.

**Table 2** Changes over time in cardiovascular and respiratory morbidity

| | Starting period | | Change from start to end period | | |
|---|---|---|---|---|---|
| | Period | Prevalence | End period | Raw change | Adj. change* (Adj. change 95% CI) |
| Blood pressure/cholesterol | | | | | |
| High blood pressure LSI | 1994–1996 | 2.7% | 2011–14 | 1.3% | 1.0% (0.4% to 1.6%) |
| Recent high blood pressure | 1994–1996 | 4.2% | 2011–14 | 5.2% | 4.8% (3.9% to 5.6%) |
| Biomarker high BP | 1994–1996 | 8.4% | 2011–14 | −4.7% | −5.0% (−5.6% to −4.5%) |
| High total cholesterol | 1994–1996 | 75.7% | 2011–14 | −16.4% | −17.6% (−19.1% to −16.1%) |
| Low HDL cholesterol | 1997–2000 | 11.8% | 2011–14 | −8.0% | −8.0% (−9.0% to −7.1%) |
| Other CVD | | | | | |
| Recent heart attack/stroke | 1994–1996 | 1.2% | 2011–14 | −0.3% | −0.4% (−0.7% to 0.0%) |
| Recent angina | 1994–1996 | 1.1% | 2011–14 | −0.4% | −0.5% (−0.8% to −0.1%) |
| IHD/stroke LSI | 1994–1996 | 1.4% | 2011–14 | −0.4% | −0.6% (−0.9% to −0.2%) |
| Heart attack symptoms | 1994–1996 | 5.5% | 2011–14 | −0.3% | −0.5% (−1.3% to 0.3%) |
| Mini stroke (TIA) symptoms | 2001–2003 | 8.1% | 2011–14 | −1.4% | −1.4% (−2.4% to −0.4%) |
| Angina symptoms | 1994–1996 | 2.3% | 2011–14 | −1.1% | −1.2% (−1.6% to −0.7%) |
| Any CVD LSI | 1994–1996 | 5.8% | 2011–14 | 1.1% | 0.6% (−0.1% to 1.4%) |
| Any recent CVD | 1994–1996 | 3.1% | 2011–14 | 0.7% | 0.5% (−0.1% to 1.2%) |
| Respiratory | | | | | |
| Lifetime diagnosed asthma | 1994–1996 | 11.2% | 2008–10 | 5.5% | 5.7% (4.5% to 6.8%) |
| Asthma LSI | 1994–1996 | 5.0% | 2011–14 | 0.7% | 0.7% (0.0% to 1.4%) |
| Breathlessness-grade 2+ | 1994–1996 | 19.7% | 2008–10 | −4.4% | −4.8% (−6.1% to −3.5%) |
| Breathlessness-grade 3 | 1994–1996 | 7.8% | 2008–10 | −1.4% | −1.6% (−2.5% to −0.8%) |
| Recent wheezing/asthma | 1994–1996 | 19.5% | 2008–10 | −1.2% | −1.2% (−2.5% to 0.1%) |
| Wheezing stopping sleep | 1994–1996 | 3.6% | 2008–10 | −0.4% | −0.5% (−1.0% to 0.1%) |
| COPD symptoms | 1994–1996 | 6.6% | 2008–10 | −1.5% | −1.6% (−2.3% to −0.8%) |

See table 1 for details on LSI.
Red text indicates negative values.
*'Adj.' = adjusted for changing age and sex distribution of the working-age population.
BP, blood pressure; CVD, cardiovascular disease; HDL, high density lipoprotein; IHD, ischaemic heart disease; LSI, longstanding illness.

To avoid a binary cut-off of statistical significance,[69] 95% CIs are used to convey precision. All analyses use weights, exclude boost samples that use different sampling methods, and adjust for the multistage clustered sample design and the stratification of the sample across survey years using the SVYSET command in Stata (although standard errors will be slightly underestimated as it is not possible to consistently adjust for sample stratification within years). For reasons of space, we are unable to discuss previous HSE studies of specific morbidity trends in the main text; these are instead described in online supplementary appendix 8.

## RESULTS
### Conditions with sharply declining mortality
We start by focussing on cardiovascular disease (CVD) and respiratory illness which have both seen large falls in mortality (by >50% and >25%, respectively, among 0–64 years old 1994–2013; online supplementary appendix 1). Changes over time in *morbidity*, however, are shown in table 2.

Looking first at high blood pressure, biomarker-measured high blood pressure has halved over two decades (similar improvements are found for the biomarkers for total and HDL cholesterol). Yet, when we look at self-reports (either people reporting this as an LSI, or in response to a direct question about having recent diagnosed high blood pressure), we see large *rises* over time. There has been an increasing diagnosis of high blood pressure and increasing prescriptions of blood pressure-lowering drugs; these may have helped reduce the underlying incidence of high blood pressure while simultaneously raising people's awareness of morbidity.

Table 2 further shows declines in several key types of CVD (heart attack, mini-stroke, angina), whether measured through people's reports of the disease itself or their reports of its symptoms. Nevertheless, the morbidity declines (8%–50%) are often not on the scale of the declines in mortality (>50%); this is likely to be because mortality declines are partly driven by improved treatment[70] which means each incident CVD case is likely to last longer.[71 72] More surprisingly, the measures of

'any reported CVD' show no improvement (with some, uncertain signs of *rises*). Looking at its sub-components (online supplementary appendix 6), this seems to be due to possible increases in diagnosed irregular heart rhythm and other heart trouble.

Finally, table 2 shows that symptoms-based measures of respiratory morbidity have improved, particularly COPD symptoms (regular cough and phlegm) and breathlessness (at both levels), and more uncertainly for recent wheezing/asthma and wheezing stopping sleep. Again, though, diagnosis-related measures of asthma—reported diagnoses, or self-reports of having asthma as a LSI—have risen, even while underlying symptomatology is improving.

Overall, table 2 illustrates how changes over time in morbidity do not necessarily follow changes in mortality. There are definite improvements in CVD risk factors and respiratory symptomatology on the scale of improvements in mortality. But the prevalence of self-reported CVD conditions such as heart attacks have only declined by a smaller amount, and recent doctor-diagnosed hypertension, any CVD, and asthma diagnoses have either stayed stable or risen.

## Conditions with claims of increasing prevalence

The previous section focused on conditions where there may be an a priori expectation that morbidity has improved (given declining mortality); in this section, we focus on three areas where there have been widespread claims of increasing prevalence—obesity, diabetes and mental health.

Looking at table 3, we do indeed confirm a large rise in obesity in HSE (an 8.0%–9.7% rise from an obesity

prevalence of 16.9% in 1994–1996). The rise in high waist-hip ratios—sometimes suggested to be a better measure of potential morbidity[73]—is even larger. This has come alongside little change in the prevalence of being *underweight* over this period.

Table 3 also confirms a large rise in diabetes. This can be seen whether diabetes is measured through people reporting diabetes as an LSI, a specific question about people currently taking medication for diabetes or via a diabetes biomarker (glycated haemoglobin). This clear rise in diabetes has occurred despite *declining* age 0–64 death rates from diabetes, which fell by more than one-third 1994–2013 (online supplemetnary appendix 1)—indeed, rising prevalence is *because of* falling mortality[74]—again demonstrating the difference between changes in mortality and morbidity.

Trends in mental health are more contentious in the wider literature (see online supplementary appendix 8), and the measures in HSE are not as strong as the more occasional Adult Psychiatric Morbidity Surveys.[75] Nevertheless, HSE offers a unique annual perspective on self-reported mental health. As we might expect from increasing treatment/diagnosis, we see a doubling in people reporting a mental health LSI. However, the symptoms-based measures show a more mixed picture:

► Neither of the measures that capture more moderate mental ill-health show rising ill-health (these are psychological distress symptoms and people reporting a feeling of anxiety/depression today, both with a relatively common prevalence of 15%–25%). If we break this down by year (see online supplemetnary

| Table 3 | Changes over time in obesity, diabetes and mental health | | | | |
|---|---|---|---|---|---|
| | **Starting period** | | **Change from start to end period** | | |
| | **Period** | **Prevalence** | **End period** | **Raw change** | **Adj. change\* (95% CI)** |
| Underweight/obesity | | | | | |
| BMI-underweight | 1994–1996 | 1.0% | 2011–2014 | –0.1% | –0.1% (–0.3% to 0.1%) |
| BMI-obese | 1994–1996 | 16.9% | 2011–2014 | 9.3% | 8.9% (8.0% to 9.7%) |
| High waist-hip ratio | 1994–1996 | 9.5% | 2011–2014 | 14.8% | 14.1% (13.0% to 15.2%) |
| Diabetes | | | | | |
| Recent diabetes | 1994–1996 | 1.2% | 2011–2014 | 2.4% | 2.2% (1.9% to 2.6%) |
| Diabetes LSI | 1994–1996 | 1.5% | 2011–2014 | 2.3% | 2.1% (1.5% to 2.6%) |
| Glycated haemoglobin | 2001–2003 | 2.7% | 2011–2014 | 2.1% | 2.1% (1.4% to 2.7%) |
| Mental health | | | | | |
| Mental health LSI | 1994–1996 | 2.1% | 2011–2014 | 2.5% | 2.4% (1.8% to 3.0%) |
| Psychological distress | 1994–1996 | 17.1% | 2011–2014 | –1.3% | –1.3% (–2.4% to –0.3%) |
| Anx./depression-moderate† | 1994–1996 | 21.9% | 2011–2014 | 0.3% | 0.1% (–1.1% to 1.3%) |
| Anx./depression-extremely† | 1994–1996 | 1.8% | 2011–2014 | 1.0% | 0.9% (0.5% to 1.3%) |

GHQ; see online supplementary appendix 5. See table 1 for details on LSI.
Red text indicates negative values.
\*'Adj.' = adjusted for changing age and sex distribution of the working-age population.
†'Anx./depression'= feeling of anxiety/depression today—see table 1.
BMI, body mass index; GHQ, General Health Questionnaire; LSI, longstanding illness.

appendix 7), we can see moderate mental ill-health symptoms fell between the mid-1990s and the mid-2000s, before rising in 2009.

► In contrast, the single measure capturing a feeling of extreme anxiety/depression today does show rising morbidity. To see if there were similar signs of rising mental ill-health at extremes in our other measure (psychological distress), we looked at a much higher GHQ threshold of 10 negative responses out of 12 questions (compared to the conventional threshold of 4). Unlike the conventional GHQ measure, this also showed an increase over time (95% CI of a 0.4% to 1.4% rise; see online supplementary appendix 6). While the GHQ is not designed to capture *severe* psychological distress in this way, others have similarly looked at moderate and extreme psychological distress using GHQ—and indeed, have found that rises in distress over time 1991–2008 are concentrated in the more extreme measure.[76]

Overall, while labelling of mental health conditions has undoubtedly risen, trends in mental health symptoms vary across measures. If we interpret higher GHQ thresholds as indicating more serious psychological distress, then we can see a consistent picture: moderate mental ill-health symptoms fell from the mid-1990s to the mid-2000s before rising around the time of the 2008 economic crisis (as we would expect),[77] whereas more extreme mental ill-health has more consistently risen.

### Activity limitations, musculoskeletal and pain

Pain/musculoskeletal conditions are a major component of working-age morbidity, yet very few previous studies show changes over time in symptomatology, and even those that exist[78] sometimes have debatable comparability.[79] Table 4 shows a fall in some—but not all—HSE measures focused on pain and musculoskeletal morbidity. Arthritis as a LSI has declined (the precision

of the estimates is greater when looking at 2008–2010 rather than 2011–2014, and shows a decline of 0.3%–1.2%). There are some (similarly uncertain) signs that other musculoskeletal LSIs have also fallen, and noticeably fewer people say that they have any pain/discomfort today, although there has been no change in people saying they have extreme pain/discomfort. The echoes a previous study that found different trends in low back pain of different levels of severity.[80]

In contrast, there has been a rise in all four activity limitations measures in HSE—although the increases are sometimes uncertain, and are smaller after adjusting for changes in age/sex structure. Moreover, the timing of the rises differ between the measures: the trend in limitations lasting at least a year shows a rise in 1994–1996 to 2001–2003, but the two measures of 'limitations today' do not, instead showing a possible slight rise in the more recent period (see online supplementary appendix 7; this difference remains if we focus on the sub-components of year-long limitations that more closely match to the 'limitations today' questions, see online supplementary appendix 6). The measures can collectively be seen as offering some, although relatively weak, evidence for an increase in activity limitations.

### Other measures

Changes over time in other measures are shown in table 5. This includes four biomarkers that are more difficult to compare directly to self-reports:

► Changes over time are available for two biomarkers of inflammation (C-reactive protein (CRP) and fibrinogen). These are associated with a number of conditions including heart disease, diabetes, cancer[81] and—in the case of CRP—even depression.[82] Table 5 shows that both biomarkers have rising morbidity from 1997 to 2000 to 2008–2010 (although for CRP,

**Table 4** Changes over time in activity limitations, pain and musculoskeletal morbidity

|  | Starting period | | Change from start to end period | | |
|---|---|---|---|---|---|
|  | Period | Prevalence | End period | Raw change | Adj. change* (95% CI) |
| Activity limitations |  |  |  |  |  |
| Problems walking about | 1994–1996 | 11.5% | 2011–2014 | 1.0% | 0.4% (−0.6% to 1.3%) |
| Any locomotor limitation | 1994–1996 | 6.8% | 2001–2003 | 1.1% | 0.9% (0.1% to 1.7%) |
| Probs. washing/dressing | 1994–1996 | 3.4% | 2011–2014 | 0.6% | 0.3% (−0.2% to 0.9%) |
| Any self-care limitation | 1994–1996 | 3.9% | 2001–2003 | 0.8% | 0.7% (0.1% to 1.3%) |
| Musculoskeletal/pain |  |  |  |  |  |
| Pain-any | 1994–1996 | 32.0% | 2011–2014 | −2.2% | −3.3% (−4.6% to −2.0%) |
| Pain-extreme | 1994–1996 | 3.0% | 2011–2014 | 0.4% | 0.2% (−0.3% to 0.7%) |
| Arthritis LSI | 1994–1996 | 5.3% | 2011–2014 | −0.3% | −0.7% (−1.4% to 0.0%) |
| Other musculoskeletal LSI | 1994–1996 | 9.7% | 2011–2014 | −0.5% | −0.8% (−1.7% to 0.1%) |

See table 1 for details on LSI.
*'Adj.'=adjusted for changing age and sex distribution of the working-age population.
LSI, longstanding illness.

**Table 5** Changes over time in other morbidity measures

| | Starting period | | Change from start to end period | | |
|---|---|---|---|---|---|
| | Period | Prevalence | End period | Raw change | Adj. change* (Adj. change 95% CI) |
| Other biomarkers | | | | | |
| Raised C-reactive protein | 1997–2000 | 21.4% | 2008–2010 | 2.1% | 1.9% (–0.7% to 4.5%) |
| Raised fibrinogen | 1997–2000 | 2.3% | 2008–2010 | 1.6% | 1.5% (0.3% to 2.6%) |
| Anaemia | 1994–1996 | 6.7% | 2008–2010 | –1.4% | –1.4% (–2.7% to –0.1%) |
| Iron deficiency | 1994–1996 | 39.9% | 2008–2010 | –12.9% | –12.5% (–14.8% to –10.2%) |
| Sensory and communication | | | | | |
| LSI eye or ear | 1994–1996 | 2.8% | 2011–2014 | –0.9% | –1.0% (–1.5% to –0.6%) |
| Hearing limitation | 1994–1996 | 4.3% | 2001–2003 | –1.5% | –1.6% (–2.1% to –1.0%) |
| Seeing limitation | 1994–1996 | 1.4% | 2001–2003 | –0.2% | –0.2% (–0.6% to 0.1%) |
| Communicating limitation | 1994–1996 | 1.0% | 2001–2003 | 0.1% | 0.1% (–0.2% to 0.4%) |

See table 1 for details on LSI.
*'Adj.'=adjusted for changing age and sex distribution of the working-age population.
LSI, longstanding illness.

the CI is wide and there is a non-negligible possibility that the change is negative).

▶ The two other biomarkers available in HSE are clearly focused on anaemia and iron deficiency. table 5 shows that both of these have declined, with particularly clear evidence for a decline in iron deficiency.

Table 5 also shows changes over time in sensory and communication-related morbidity. This shows a fall in eye/ear conditions (1994–1996 to 2011–2014) as well as hearing limitations in the earlier period (1994–1996 to 2001–2003), but no change in people having difficulty communicating with others.

## DISCUSSION

Despite considerable evidence on morbidity trends among older people, there are few published studies on changes in morbidity among the working-age population, particularly outside the USA. In this paper, we have analysed changes over time in working-age morbidity in England 1994–2014 using a high-quality repeated cross-sectional study. We see improvements in cardiovascular morbidity, respiratory morbidity and anaemia, but deteriorating obesity, diabetes, some biomarkers (fibrinogen and possibly also CRP) and feelings of extreme anxiety/depression. We see little systematic change over time in more common mental ill-health or musculoskeletal conditions, pain/mobility and self-care limitations. Symptomatology and chronic disease diagnoses also often go in different directions—chronic disease diagnoses have sometimes stayed stable or even risen at the same time that underlying symptomatology has declined (such as for mental health conditions, asthma, hypertension and CVD as a whole), mirroring findings at older ages.[3]

Our analysis has several strengths. We include every morbidity measure for which consistent changes can be constructed, including chronic disease, functioning and symptomatology, and biomarkers. We use a single survey series collected by a single survey organisation; exclude under-25s for whom comparability of survey coverage is unlikely; and construct new non-response weights. Nevertheless, we must note three limitations. First, response rates for each stage of the HSE have declined over time (see online supplementary appendix 3), and while we create new non-response weights covering the entire period, it is still possible that socioeconomically disadvantaged people (within any age-sex-region group) have become less likely to respond—and as they tend to be in worse health, this could mask deteriorating morbidity. Second, even if non-response biases have not changed, it is possible that people respond differently over time even to identical questions. Third, there are several dimensions of morbidity for which there is little comparable data in HSE. This includes several areas in which morbidity among the working-age population seems to be rising, including *inter alia* cognitive complaints,[83] allergic disorders[84] and liver cirrhosis (see online supplementary appendix 1), as well as some areas in which morbidity seems likely to have fallen, such as chronic kidney disease.[85]

It is clear that there are different trends in different dimensions of morbidity—but for policy-makers, this leaves the question of whether working-age morbidity as a whole is unchanged (H2), getting better (H1) or getting worse (H3), to the extent that it makes sense to place health on a unidimensional scale. While we cannot create a single morbidity index here, online supplementary appendix 9 shows the association of each measure with bad general self-rated health (net of age, gender and education). This shows little systematic trend for falling morbidity to be seen in the measures that predict health the most (indeed, the evidence weakly points in the other direction, towards rising morbidity). This provides greater support for H2 than H1 or H3, mirroring evidence from

the Global Burden of Disease study (see online supplementary appendix 9).

In conclusion, despite considerable falls in working-age mortality and gains in life expectancy—and the ensuing expectations of social security policy-makers for improving morbidity—there is no evidence of systematic improvement in overall working-age morbidity in England from 1994 to 2014. However, two pieces of further research could strengthen this evidence base. First, the ideal measures for analysing changes in morbidity are functional limitations measures which are included in the HSE from 1996. However, these were last asked to the working-age population in 2001, and it is a priority to repeat these measures in future years of HSE. Second, there is a surprising paucity of studies looking at the changing morbidity of the working-age population outside the USA. Given their importance in public debate—particularly in discussions of retirement ages and disability benefits—we hope that other authors will repeat and extend our analyses here, including disaggregating these changes across different regions and sociodemographic groups.

**Correction notice** This article has been corrected since it was first published. Data in the table 2-5 has been corrected.

**Acknowledgements** Many thanks to Clare Bambra for comments, and to Mariska van der Horst for research assistance; neither should be held responsible for the analysis or interpretation of the paper itself.

**Contributors** BBG was responsible for the design, data preparation, analysis and reporting of the study.

**Funding** This work was supported by the Economic and Social Research Council (grant number ES/K009583/1).

**Competing interests** No.

**Patient consent for publication** Not required.

**Provenance and peer review** Not commissioned; externally peer reviewed.

**Data availability statement** Data are available in a public, open access repository. The Health Survey for England 1994-2014 are available for free to registered users at the UK Data Service - see https://beta.ukdataservice.ac.uk/datacatalogue/series/series?id=2000021#!/abstract. There are no conditions for re-use for non-commercial applications of the data. The statistical code enabling replication using publicly available data is available from OSF (Morbidity in England 1994-2014 2019, available from: http://osf.io/dy6sv) and www.benbgeiger.co.uk.

**ORCID iD**
Ben Baumberg Geiger http://orcid.org/0000-0003-0341-3532

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
