## [Reviewer comments · BMJ Open]

ARTICLE DETAILS

TITLE (PROVISIONAL)	Has working-age morbidity been declining? Changes over time in survey measures of general health, chronic diseases, symptoms and biomarkers in England 1994-2014
AUTHORS	Geiger, Ben

VERSION 1 – REVIEW

REVIEWER	Sarah Cook London School of Hygiene & Tropical Medicine
REVIEW RETURNED	28-Jun-2019

GENERAL COMMENTS	This is an interesting and well written paper looking at changes in morbidity over time within the Health Survey for England using a variety of different types of measure used consistently across waves. The comparison of different types of data used and their comparison is well described and gives some insight into the differences obtained when using self-reported diagnoses compared to physical measures. Issues for consideration: 1) The analysis section of the methods needs more detail about what was done including explicitly stating the models that were fitted.a. From the text and models presented it appeared that logistic regression models were used with time (grouped in 4 year categories) as the exposure treated as continuous and each morbidity (binary) as an outcome with adjustment for age and sex. However it may have been a model comparing the earliest and latest time period. This is not defined very clearly and could be made more explicit in the text.b. If logistic regression models were used it is not clear why % changes are presented – from a logistic model measure of effect would be odds ratiosc. “All analyses... adjust for the clustered nature of the main sample” Please explain how adjustment for clustering was carried out.2) Was a linear effect of time assumed? Was any investigation of this done? The appendix include analyses for each year which is useful but it would be of benefit to investigate formally whether there is departure from linearity and whether presenting tests for trend is appropriate.3) Models are adjusted only for age and sex. Reasons for not including other confounders should be justified. It is reasonable that other factors were considered as mediators rather than confounders and not included however given the changes in
--

	response level over time other factors such as educational distribution in the sample may have changed over time which may not necessarily reflect population changes but rather changes in constitution of the health survey for England. This may have been accounted for with the survey weights but it did not seem to me from the information presented that it had been. Some investigation and discussion of changes in survey composition over time (due to the changing response rates) and potential impact on the findings would be of benefit. 4) While models were adjusted for sex – consideration of whether the associations were the same in men and women (interaction by sex) would have been interesting. Reporting of symptoms is often higher in women regardless of clinical pathology. Stratifying by sex as a further analysis would strengthen the manuscript (could be additional supplementary material). 5) Throughout “incidence” is used when author is actually talking about prevalence. 6) Presentation of the tables – the presentation of findings in Tables 2,3,4 needs revision as the tables currently reflect the uncertainty about what was done (see comment 1a). There is a column heading “trend” but it is not clear what this definitely represents – is this a linear trend with time category?; effect estimates are % but in methods it says logistic model was used (in which case odds ratios should be presented); Column heading “Incidence” when from methods and text in results it sounds like change in prevalence was assessed.
--	---

REVIEWER	Joel Coste Paris Descartes University, France
REVIEW RETURNED	10-Jul-2019

GENERAL COMMENTS	This paper investigated trends in morbidity in England between 1994 and 2014 using 39 measures from the Health Survey for England that are comparable over time. The authors observed improvements in cardiovascular morbidity, respiratory morbidity and anaemia, but deteriorating obesity, diabetes, some biomarkers (fibrinogen, CRP) and mental ill-health at the highest levels. However, the authors found symptomology measures and chronic disease diagnoses may go in different directions. General comments The manuscript is generally well written but the authors should more clearly state 1) the objectives of the study and hypotheses; 2) how the methods and results align with those objectives; and 3) how the different measures may provide converging, diverging or complementary evidence. The respective place of the various measures of global/focal health/disability should be better and earlier defined. As the studied population is working age, one would expect more attention to be paid to work status and potential influence of macrosocioeconomic factors, especially the 2008 crisis. It is a bit surprising that non-linear trends have not been investigated eg. using fractional polynomials. Specific comments
---

	1. Describe and discuss further “changing non-response biases”. This is a key problem in this type of study. 2. Discuss further why people may respond differently over time to identical questions and relate this problem to that often referred to in the literature as “response shift”. 3. There is some confusion about “Effect on bad health” in Appendix 2. Please explain how it has been calculated, the footnote of Figure 1 is rather unclear.
--	---

REVIEWER	Jan R. Boehnke University of Dundee, UK
REVIEW RETURNED	22-Jul-2019

GENERAL COMMENTS	The manuscript "Has working-age morbidity been declining? Trends in general health, chronic diseases, symptoms and biomarkers in England 1994-2014" (bmjopen-2019-032378) presents a detailed descriptive analysis of prevalences of health and illness markers in the UK working age population. The paper presents adjusted and unadjusted estimates of prevalences in 39 (including appendices even more) markers of health and illness that have been collected in the Health Survey for England across a roughly 20 year period. The introduction gives a brief, but comprehensive lead-in as well as strong motivation for the piece of research; the choices and methodological considerations are spelled out in detail; and the produced code/data-set is likely to be an important resource for other researchers. The writing is dense with a lot of appendices, but I could not find a single piece of irrelevant information. Although I have a number of questions and suggestions in the following, this is likely to be the most exciting piece of research I have reviewed this year so far. MAJOR POINTS 1) NON-RESPONSE WEIGHTS 1a) Neither the information in Appendix I nor in Appendix III is enough to construct the non-response weights. This could probably be sorted by providing the code for the analyses as promised, but the description could also be more detailed. 1b) A minor point, but related to this: I think in Appendix I a lot of references should actually point to Appendix III? 1c) Another minor point, but related: The code should not be hosted on the author's web page, since these are subject to frequent changes. Since many different repositories exist (OSF, github,...) that are for exactly this purpose, the code can easily be shared over one of those pages. [I do not think this is an issue since the author already indicated that it will be shared] 1d) Without sharing the code with the review it is really difficult to evaluate (a) whether all code and data have been shared and (b) how appropriate the applied methods were. [I do not think this is an issue since the author already indicated that it will be shared]
---

General and maybe helpful reminder for 1c-1d: BMJOpen review is not blind, so sharing the code is possible through any repository (most would even allow an anonymous link anyway).

2) ADJUSTED ANALYSES

2a) I appreciate that a lot of questions can be thrown at any adjustment. The current analyses were nevertheless only adjusted for age and sex. We nevertheless know that the socio-economic gradient is very strong in the UK for the research target in question, so that changes in the composition of this sample with regards to this could have an important impact.

2b) Also regional markers are available to control to some degree for region. It is well-established that mortality differs markedly across regions, why was it not adjusted for?

2c) Additionally, it is unclear what "adjustment" really means. Were again weights calculated and averages based on double weighting presented? Or were regression adjustments made? (how were the prevalences derived then or are these actually margins?) The statement "(sex/age-adjusted models show average marginal effects following a logistic regression)." does not really provide enough information.

2d) How were the variables for the selection weights chosen?

3) "TRENDS"

I think overall the analysis cannot be phrased as focusing on trends. If I understand the analyses and all tables correctly, it is a two-time point analysis (albeit both time points represent aggregated data from longer periods). The paper describes the methodology and choices to derive these periods in a lot of detail, so I think it is generally clear. But with a number of additional discussion points especially in the appendices it becomes clear that the analysis is not fine-grained enough to really represent "trends" (e.g. increase, peak and decrease in mental health problems around and potentially due to the financial crisis). Therefore I think the term "trend" should be removed from the paper.

I still think the presented results are immensely rich and informative, but they do not represent trends.

4) THE GHQ-12

The question what the GHQ-12 exactly measures is contentious and given the plethora of available analyses and suggested structures, difficult to pin down. Nevertheless, I would suggest not to use "anxiety/depression" as a description since this is suggestive of the psychiatric categories. This is disadvantageous, because (i) although developed for screening, the GHQ was never intended as a marker of specific diagnostic categories and does not function as such (e.g., Goldberg et al., 1998, Psych Medicine, 915-921; Goldberg & Hillier, 1979, Psychol Medicine, 139-145); and (ii) for the assessment of diagnostic categories at least a

interviewer checklist, but better an actual clinical interview should have been conducted.

Several descriptive labels for the measured dimension exist, to give some examples:

In the early days/1980ies around (levels of) 'non-psychotic mental illness' or Goldberg & Hillier writing about the lowest common multiple of symptoms that can be encountered in a variety of mental illnesses. Today, we have proposed language around 'psychological distress'/'common mental distress' (e.g., Stochl et al., 2016a, BMC Med Res Meth, 16:58; 2016b, Soc Psychiatry Psychiatr Epidemiology, 895-906) and other suggest 'psychological morbidity' (Smith et al., 2013, Qual Life Res, 145-152). Given the vast amount of literature around the scale (e.g., see introduction in Gnamb & Staufenbiel, 2018, Health Psychology Review, 12, 179-194), I am sure that the author can find terminology that avoids overlap with clinical diagnostic categories.

5) Inflammatory markers

Page 21: CRP and fibrinogen are not commonly used measures of heart disease risk (for that other biomarkers would be needed, e.g. Troponin). I would suggest to turn the presentation around and state that both are generic markers of inflammatory processes and have been linked to (and then the diseases and disorders with the references as presented in the paragraph).

MINOR

6) Author statement unclear: "A research assistant (Mariska van der Horst) briefly helped with some of the data preparation." The named researcher is not an author and would not qualify according to ICMJE criteria. The input is noted appropriately under "Acknowledgements".

7) I think generally the manuscript could do with a read through for adjectives and comparatives. They can largely be removed. It is a detailed descriptive paper. It is for the readers to assess whether an increase or decline of something is particularly "sharp" or "marked". Without clear guidelines on something is interpreted as such, it is subjective and there is no need to qualify the reported numbers further.

8) page 6, "(which are unreliable, as we explain below)" This is a throwaway statement here and should be removed. The discussion further down in the manuscript should be enough.

9) page 7: "which provide further insight into whether reported changes are simply reporting changes"
This is unclear to me and could maybe rephrased?

10) Appendix I: "Because of the high level of item non-response for BMI, a non-response weight was created to try to correct for any biases that this introduces. This followed the identical procedure outlined in Appendix 1 for creating non-response weights for the nurse visit, blood sample etc."

This reference is slightly confusing: (i) this is Appendix I and at this stage no procedure has been described; (ii) even in later parts of

	Appendix I no procedure for determining the non-response weights has been described. Such references are made multiple times in Appendix I, but no procedure to determine the weights is described at any point. Later I noticed: Maybe these references should lead to Appendix 3? And once the code is provided this should be review-able (at the moment no formulae etc presented, so unclear what was exactly done). 11) Appendix 4: "For several of the general health measures, there is more evidence of change over this period..." 'More' compared to what? 12) Appendix 4: "As an aside, UK Government publications have made claims based on healthy/disability-free life expectancy, most recently to argue that morbidity has been deteriorating." Please provide references and point clearly to those claims. Otherwise this reads like opinion/conjecture. 12) Page 19 & Appendix 8: Important information is discussed here. With view to the trends in mental health it is maybe worth noting that the rise and peak of mental health problems may be related to the financial crisis and this has been widely discussed in the literature and may be worth noting. Our paper on this topic (Kronenberg & Boehnke, 2019, Econ Hum Biol. 2019 May;33:193-200) is maybe a bit too much work place-focused, but our introduction might contain one or two references that can provide some context for this trend. 13) Page 22, "but deteriorating obesity, diabetes, some biomarkers (fibrinogen and possibly also CRP) and mental ill-health at the highest levels" Unclear what "we see deteriorating x at highest levels" means 14) Page 23, typo: symptom_a_tology?
--	--

VERSION 1 – AUTHOR RESPONSE

Overview

R1: This is an interesting and well written paper looking at changes in morbidity over time within the Health Survey for England using a variety of different types of measure used consistently across waves. The comparison of different types of data used and their comparison is well described and gives some insight into the differences obtained when using self-reported diagnoses compared to physical measures.

R3: The introduction gives a brief, but comprehensive lead-in as well as strong motivation for the piece of research; the choices and methodological considerations are spelled out in detail; and the produced code/data-set is likely to be an important resource for other researchers. The writing is dense with a lot of appendices, but I could not find a single piece of irrelevant information. Although I have a number of questions and suggestions in the following, this is likely to be the most exciting piece of research I have reviewed this year so far.

Response: Many thanks to all three reviewers for their positive words, as well as the constructive criticism and suggestions for improvement that I respond to below – your time is enormously appreciated, and I have tried to match this by putting time into this response.

Framing of the study

R2: The manuscript is generally well written but the authors should more clearly state 1) the objectives of the study and hypotheses; 2) how the methods and results align with those objectives; 3) how the different measures may provide converging, diverging or complementary evidence. The respective place of the various measures of global/focal health/disability should be better and earlier defined.

Response: this point is well-taken – particularly around the discussion of the different measures, and the extent to which each of these provides helpful evidence in responding to our overall question. I have particularly extended the section on ‘Measures’, and have also made changes to the Introduction and Discussion too.

It is worth noting that I do not have formal hypotheses in this paper – it is a descriptive paper, rather than one testing a theoretical account. I also do not perform conventional statistical significance testing for reasons given in the paper itself, and deliberately avoid binary rejection/acceptance of the null hypothesis. There are many defensible responses to the current challenges in significance testing, but hopefully you agree that this is one of several reasonable responses.

R2: Discuss further why people may respond differently over time to identical questions and relate this problem to that often referred to in the literature as “response shift”.

Response: again the point is well-taken; this issue is now covered in the same section as the preceding point.

Methods of analysing change over time

R1: The analysis section of the methods needs more detail about what was done including explicitly stating the models that were fitted... From the text and models presented it appeared that logistic regression models were used with time (grouped in 4 year categories) as the exposure treated as continuous and each morbidity (binary) as an outcome with adjustment for age and sex. However it may have been a model comparing the earliest and latest time period. This is not defined very clearly and could be made more explicit in the text.

Response: My apologies for not including the regression equation in the paper; this has now been included. [Apologies too that this is not included in editable equation format; my version of Word kept corrupting when these were included, which sometimes happens when there are lots of tracked changes as well as equations. But hopefully you will be able to see the equations as an image embedded within the text].

R1: Was a linear effect of time assumed? Was any investigation of this done? The appendix include analyses for each year which is useful but it would be of benefit to investigate formally whether there is departure from linearity and whether presenting tests for trend is appropriate.

Response: Apologies for not making this clear in the original paper. Rather than treating year or period as a continuous variable, I used dummy variables for each period. Given our aims of informing policy debates, this has three advantages: a discrete change is simple to interpret; it is compatible with the different start/end year available for different measures; and it does not require any assumptions about the functional form of trends. My expectation was that many of the trends would be non-linear, particularly for indicators that are more closely relating to non-linear economic cycles or policy shifts (for evidence that this is indeed the case, see e.g. the detailed discussion in the paper/appendices on the mental health measures).

It is worth adding that several of the papers looking at trends in morbidity that I review use a similar approach (e.g. Freedman et al 2007 and Freedman et al 2014). I have now explained and justified the approach explicitly in the paper – many thanks for prompting me to do this.

R1: Presentation of the tables – the presentation of findings in Tables 2,3,4 needs revision as the tables currently reflect the uncertainty about what was done (see comment 1a, on whether linear trends or discrete changes over time were used). There is a column heading “trend” but it is not clear what this definitely represents – is this a linear trend with time category?

Response: Apologies that this was not clear, and many thanks for suggesting that this is corrected – all of the tables have now been changed.

R2: It is a bit surprising that non-linear trends have not been investigated eg. using fractional polynomials.

Response: My apologies that the method of assessing change over time was not clear in the original paper – this has now been corrected (see preceding comments). Given that I am looking at discrete changes between periods, issues of linearity and non-linearity do not arise; indeed, the likelihood of non-linearity is one of the reasons why discrete changes between periods are used.

(As an aside: I have used fractional polynomials elsewhere, and they are indeed an excellent suggestion if I wanted to use continuous measures of time but avoid the assumption of linearity. However, while they are easy to use (using the MFP package in Stata), they are difficult to explain; and given one of my aims is to provide clear evidence to underpin policy, they are not appropriate here).

R3: I think overall the analysis cannot be phrased as focusing on trends. If I understand the analyses and all tables correctly, it is a two-time point analysis (albeit both time points represent aggregated data from longer periods). The paper describes the methodology and choices to derive these periods in a lot of detail, so I think it is generally clear. But with a number of additional discussion points especially in the appendices it becomes clear that the analysis is not fine-grained enough to really represent “trends” (e.g. increase, peak and decrease in mental health problems around and potentially due to the financial crisis). Therefore I think the term “trend” should be removed from the paper. I still think the presented results are immensely rich and informative, but they do not represent trends.

Response: This is an interesting comment. My understanding is that the reviewer defines ‘trend’ as referring to the detailed course of an indicator over time, rather than merely whether it is higher at the end than the start. This is a reasonable definition, although it is worth noting that dictionary definitions are ambiguous as to whether the latter should be excluded (e.g. Oxford define it as, “A general direction in which something is developing or changing”; Collins as “a change or development towards something new or different”).

It could be argued that my paper does cover trends even in the former sense. The models underlying the main paper include dummies for the full set of time periods; in Web Appendix 7 I show detailed year-by-year trends; and where relevant (notably for mental health), I discuss the detailed course of change over time.

Nevertheless, overall I agree with the reviewer that the paper focuses primarily on change between a starting period and an end period. I have therefore changed the terminology of the paper throughout (except when discussing bona fide trends under mental health), including in the title of the paper.

Confounders

R1: Models are adjusted only for age and sex. Reasons for not including other confounders should be justified. It is reasonable that other factors were considered as mediators rather than confounders and not included... [continued under Survey weights below].

R3: Also regional markers are available to control to some degree for region. it is well-established that mortality differs markedly across regions, why was it not adjusted for?

Response: This paper is focused on describing changes in morbidity, rather than on explaining them. In general, changing age and sex distributions are seen being methodological artefacts rather than as explanations for a 'real' trend – hence the conventional practice of age-sex standardisation when comparing mortality rates across time/place. In contrast, other sociodemographic changes such as the changing regional balance of population, changing qualifications and changing work patterns are seen as potential explanations of changing morbidity/mortality, rather than as artefactual.

As such, this paper does not control for region, education or employment status; but the dataset that I have created does allow these to be investigated by other researchers. Thanks for drawing my attention to the lack of discussion of this in the original version; I have made this much clearer in the revised paper.

R1: While models were adjusted for sex – consideration of whether the associations were the same in men and women (interaction by sex) would have been interesting. Reporting of symptoms is often higher in women regardless of clinical pathology. Stratifying by sex as a further analysis would strengthen the manuscript (could be additional supplementary material).

Response: this is indeed interesting. However, I feel that the paper is already very full given the large number of outcome variables and extensive documentation, and that including yet more supplementary materials could be overwhelming.

However, given that you have an interest in this, I have run these analyses specifically for you. The full results are given at the end of this document, but in summary:

- *There are some morbidity domains for which there are considerable and systematic differences in changes over time between men and women. In particular, all of the blood pressure and obesity indicators show different changes over time by gender, as does iron deficiency and some of the diabetes indicators. The directions of these differences are not consistent:*
 - *Women show more positive morbidity trends than men for some of these (diagnosed high blood pressure, low HDL cholesterol, high BMI, diabetes, iron deficiency);*
 - *Men show more positive trends than women for others (biomarker high blood pressure, low HDL cholesterol, high waist-hip ratio).*
- *However, for most indicators there are no systematic gender differences. This includes CVD (excluding blood pressure), respiratory morbidity, mental health, and most of the activity limitations/musculoskeletal indicators (barring 'other musculoskeletal LSI').*

It is worth noting that these differences are complex to describe and defy easy interpretation – which makes it even harder to adequately include these in the paper without overwhelming the reader. However, hopefully the results are interesting to you, and do let me know if you would like to explore these further (the code constructing the dataset is also now publicly available at the request of Reviewer #3, if you want to explore these further! The syntax for the stratified analysis is included within this.)

R2: As the studied population is working age, one would expect more attention to be paid to work status and potential influence of macrosocioeconomic factors, especially the 2008 crisis.

Response: the 2008 crisis is clearly a major influence on working-age morbidity, as are macroeconomic factors (and other societal factors such as public spending and social protection) more generally. However, this paper is focused solely on describing morbidity trends, rather than explaining them, as I now make clearer in the paper itself – and it is therefore beyond the scope of the paper (and the word limit) to conduct further analyses looking at the role of the 2008 crisis. There are however three further issues here to comment on further.

Firstly, the morbidity effects of the 2008 crisis are likely to be complex. As Burgard et al point out (<https://doi.org/10.1177%2F0002716213500212>), there are two contrasting literatures on the health effects of recessions. One focuses on unemployment and financial pressures as risk factors for ill-health. The other, in contrast, focuses on the net effects of economic recessions at the aggregate level – and this finds that economic recessions are BENEFICIAL for health, on average (for a recent summary by a leading contributor to this debate, see Ruhm 2016 <https://doi.org/10.1002/hec.3373>). This is because the increase in e.g. suicide is usually outweighed by reduced consumption of unhealthy products and lower road traffic mortalities. There are further complexities here – not all recessions are equal, and the effect of the recent Icelandic recession is completely different to the economic crash in post-Soviet states; and whether these findings that are predominantly on mortality will carry over to morbidity is unclear. It is therefore difficult to speak about the effect of macroeconomic factors in brief; they call for fuller, more nuanced treatment.

Secondly, the dataset that I have constructed will allow others to investigate the role of macroeconomic factors in ill-health. This is partly because it includes a large range of morbidity indicators over an extensive period (and also includes measures of region, to allow for sub-regional analysis of economic indicators), but also because it allows analysis on the individual level as it includes consistent measures of employment and education. I am therefore working with collaborators that specialise in this area to do further work on this.

Third, there is one area where there is a relatively unambiguous role of the economic crisis in morbidity – and that is for mental health. In response to the reviewer comment, I have therefore briefly mentioned the role of the crisis in discussing mental health morbidity trends; see also the following response to R3.

R3: Page 19 & Appendix 8: Important information is discussed here. With view to the trends in mental health it is maybe worth noting that the rise and peak of mental health problems may be related to the financial crisis and this has been widely discussed in the literature and may be worth noting. Our paper on this topic (Kronenberg & Boehnke, 2019, Econ Hum Biol. 2019 May;33:193-200) is maybe a bit too much work place-focused, but our introduction might contain one or two references that can provide some context for this trend.

Response: many thanks for this comment – I had not yet seen your recent paper on this, which is very interesting! As you said, it is probably more suitable to cite some of the general papers that you cite rather than your paper itself; but I have added a relevant reference amid a short discussion of the role of the crisis in mental health trends (see also the preceding response to R2).

Survey weights / non-response

R3: How were the variables for the selection weights chosen?

Response: There are three key challenges in creating consistent non-response weights over an extended period: (i) creating consistent variables in the dataset (e.g. a consistently coded education variable), (ii) creating a consistent population baseline to weight against (e.g. to

know what the educational levels in the population as a whole are), and (iii) ensuring that these two measures match one another. Age-sex-region weighting is relatively straightforward over long periods as the variables can be made consistent over time within HSE (much as this required an additional dataset to be obtained from NatCen; see Appendix 3), and there are comparable, publicly available age-sex-region baseline figures from the population available from the Office of National Statistics via *nomis*.

It is probably not completely impossible to weight HSE by further variables, but this task would be an order of magnitude more resource-intensive than the reweighting I have done here – which itself was an effort that no previous HSE analyst has done. I have drawn attention to the limitations of this (see the following response), but my view is that despite these imperfections, my new weights are a helpful contribution to future researchers, as well as allowing a more robust analysis of long-term trends over time than has previously been possible in the absence of weights pre-2003.

It should also be noted that for the second-stage non-response adjustment (e.g. people who complete the initial questionnaire but not the nurse visit), I adjust for a greater number of variables as the challenges here are lessened: this includes age and gender, qualifications, household type, employment status, smoking, and self-reported general health (see Appendix 3).

R1: ... given the changes in response level over time other factors [*than age/sex*] such as educational distribution in the sample may have changed over time which may not necessarily reflect population changes but rather changes in constitution of the Health Survey for England. This may have been accounted for with the survey weights but it did not seem to me from the information presented that it had been. Some investigation and discussion of changes in survey composition over time (due to the changing response rates) and potential impact on the findings would be of benefit.

R3: I appreciate that a lot of questions can be thrown at any adjustment. The current analyses were nevertheless only adjusted for age and sex. We nevertheless know that the socio-economic gradient is very strong in the UK for the research target in question, so that changes in the composition of this sample with regards to this could have an important impact.

R2: Describe and discuss further “changing non-response biases”. This is a key problem in this type of study.

Response: these comments are well-taken – I am not able to adjust for changing unit non-response biases beyond those that are related to age, sex and region, and it is impossible to rule out the possibility that there are changing non-response biases with respect to education.

It is not easy to investigate the extent of changing non-response biases with respect to education, due to the three challenges identified in the preceding response to reviewers. I have overcome the first challenge – I have created a consistent measure of education in HSE over time. However, we struggle much more to find a gold-standard population level measure of qualification levels:

- *Conventionally Government statistics use the Labour Force Survey / Annual Population Survey (e.g. Table 3.4 of Education and training statistics for the UK: 2018 at <https://www.gov.uk/government/statistics/education-and-training-statistics-for-the-uk-2018>, or the 2016 publication Qualifications in the population at <https://www.gov.uk/government/statistical-data-sets/fe-data-library-qualifications-in-the-population-based-on-the-labour-force-survey>)*
- *However, this is obviously just another sample survey, subject to the same issues of changing non-response biases as other surveys. Indeed, as I have shown elsewhere (Social Science & Medicine 2015, <http://dx.doi.org/10.1016/j.socscimed.2015.07.012>),*

there are various ways in which the LFS is less robust than the HSE, e.g. the majority of the sample is based on follow-up interviews from an initial interview, which is therefore further subject to attrition bias between waves; the follow-up interviews use telephone interviews whereas the initial interviews are face-to-face, creating survey mode differences between waves etc).

- Nevertheless, I can compare HSE to LFS. Using the same methods as the aforementioned Social Science & Medicine paper (which creates a bespoke comparable education variable between HSE & LFS), I find that the education trends in HSE and LFS are similar. For example, looking at degree-level / NVQ4-level qualifications (the highest category), we see a 10 percentage point rise 1998-2010 in both GHS and LFS (whether using weighted or unweighted data):

	Weighted		Unweighted	
	LFS	HSE	LFS	HSE
1998	13.4%	16.1%	13.6%	15.6%
1999	14.4%	17.2%	14.4%	16.4%
2000	15.0%	19.2%	15.0%	17.8%
2001	16.0%	19.2%	15.9%	18.3%
2002	16.0%	19.6%	16.1%	19.0%
2003	17.2%	20.8%	17.3%	20.2%
2004	17.0%	22.3%	17.2%	21.5%
2005	17.8%	22.5%	17.9%	21.5%
2006	20.1%	23.8%	20.2%	23.4%
2007	20.7%	24.3%	21.0%	23.8%
2008	20.6%	24.5%	21.0%	23.7%
2009	21.9%	25.9%	22.5%	24.5%
2010	23.9%	26.6%	24.3%	26.0%

This offers some reassurance that there are no particular issues re changing non-response biases with respect to the education profile of HSE, at least compared to the LFS. We can never rule out changing non-response biases over time, but this additional analysis shows that there is nothing in the data that suggests that we should be unusually worried.

I have also slightly extended the discussion of changing non-response biases in the text (while staying within the word limit), as suggested.

R3: Appendix I: "Because of the high level of item non-response for BMI, a non-response weight was created to try to correct for any biases that this introduces. This followed the identical procedure outlined in Appendix 1 for creating non-response weights for the nurse visit, blood sample etc." This reference is slightly confusing: (i) this is Appendix I and at this stage no procedure has been described; (ii) even in later parts of Appendix I no procedure for determining the non-response weights has been described. Such references are made multiple times in Appendix I, but no procedure to determine the weights is described at any point. Later I noticed: Maybe these references should lead to Appendix 3?

Response: I appreciate the reviewer reading through the appendices in such detail. In response:

- I have corrected the reference to the Appendix to refer to the correct Appendix (Appendix 3) – and more generally, I have checked the main text and appendices to ensure that references to appendix numbers are correct. (I was required during the submission

process to re-number the appendices in the order in which they appear in the text, at which point occasional numbering errors were introduced).

- *Appendix 1 now does not refer to weights until the end of the appendix, where it refers to Appendix 3.*

See also the response to the following comment.

R3: Neither the information in Appendix I nor in Appendix III is enough to construct the non-response weights. This could probably be sorted by providing the code for the analyses as promised, but the description could also be more detailed.

[And once the code is provided this should be review-able (at the moment no formulae etc presented, so unclear what was exactly done).]

Response: Appendix 3 has now been slightly restructured and clarified; hopefully it is now clear how the weights were constructed (see also response to the comment about supplying the code below).

Methodological detail

R1: The analysis section of the methods needs more detail about what was done including explicitly stating the models that were fitted... If logistic regression models were used it is not clear why % changes are presented – from a logistic model measure of effect would be odds ratios

R1: Presentation of the tables: ...effect estimates are % but in methods it says logistic model was used (in which case odds ratios should be presented)

R3: Additionally, it is unclear what "adjustment" really means. Were again weights calculated and averages based on double weighting presented? Or were regression adjustments made? (how are the prevalences derived then or are these actually margins?) The statement "(sex/age-adjusted models show average marginal effects following a logistic regression)." does not really provide enough information.

Response to all of these comments: apologies that the original analysis section was too brief. It now provides much more detail on the model that was fitted, and how the adjustment for age and sex takes place (see also the separate section on 'Confounders').

In terms of my use of % changes (rather than odds ratios) – these have two advantages:

- *Firstly, these are simple to understand – I realise that there is a live debate on the interpretability of odds ratios, but my view is that these have no easy real-world interpretation. (Odds ratios approximate relative risk for rare outcomes, but this approximation becomes increasingly poor for outcomes that are more common – see e.g. <https://www.ncbi.nlm.nih.gov/pmc/articles/PMC4640017/>).*
- *Secondly, and more importantly, odds ratios are not fully comparable across different models, and cannot therefore underpin our comparison between indicators of changes over time (I usually reference Carina Mood's influential 2010 paper in the European Sociological Review which kick-started the current debate about this, but Winship & Mare wrote about this in the American Sociological Review back in 1984). This is increasingly widely-recognised, and has even been noted in the existing literature on this topic: Martin & Schoeni 2015 note that "care must be taken in comparing ORs across logistic models, since the variance of the outcome (a component of the OR estimation) changes as variables are added to models (unlike the case for OLS regression in which the variance is fixed)."*

Average marginal effects are one of Mood's recommended solutions to this issue, as these are fully comparable across models – and are also more easily interpretable. They are not without their downsides (they are not on the scale on which the underlying model is constructed, and they are computationally more intensive), but my view is that they are the best choice given the aims of the paper. I have now justified this choice in the paper.

R1: The analysis section of the methods needs more detail about what was done including explicitly stating the models that were fitted... All analyses... adjust for the clustered nature of the main sample" Please explain how adjustment for clustering was carried out.

Response: apologies for not having made this clearer in the original paper – I have now clarified in the text that I adjust for the multistage clustered sample design and the stratification of the sample across survey years using the SVYSET command in Stata (although standard errors will be slightly underestimated as it is not possible to consistently adjust for sample stratification within years). In case you have any familiarity with this command, the syntax for this adjustment is

```
svyset _uniquePSU [pweight=`weight'], strata(_svyyear)
```

...where _uniquePSU is the primary sampling unit, _svyyear is the survey year, and `weight' reflects whichever weight is relevant for a particular variable (this is different for a measure taken from the blood sample than for a measure within the self-completion questionnaire; see Appendix 2). Further detail on the calculation using the SVYSET command is given at <https://www.stata.com/manuals/svyvarianceestimation.pdf>

R2: There is some confusion about "Effect on bad health" in Appendix 2. Please explain how it has been calculated, the footnote of Figure 1 is rather unclear.

Response: apologies that this was unclear; I have added more detail to the description in the appendix.

Labelling of morbidity indicators

R3: The question what the GHQ-12 exactly measures is contentious and given the plethora of available analyses and suggested structures, difficult to pin down. Nevertheless, I would suggest not to use "anxiety/depression" as a description since this is suggestive of the psychiatric categories. This is disadvantageous, because (i) although developed for screening, the GHQ was never intended as a marker of specific diagnostic categories and does not function as such (e.g., Goldberg et al., 1998, Psych Medicine, 915-921; Goldberg & Hillier, 1979, Psychol Medicine, 139-145); and (ii) for the assessment of diagnostic categories at least an interviewer checklist, but better an actual clinical interview should have been conducted.

Several descriptive labels for the measured dimension exist, to give some examples:

In the early days/1980ies around (levels of) 'non-psychotic mental illness' or Goldberg & Hillier writing about the lowest common multiple of symptoms that can be encountered in a variety of mental illnesses.

Today, we have proposed language around 'psychological distress'/'common mental distress' (e.g., Stochl et al., 2016a, BMC Med Res Meth, 16:58; 2016b, Soc Psychiatry Psychiatr Epidemiology, 895-906) and other suggest 'psychological morbidity' (Smith et al., 2013, Qual Life Res, 145-152).

Given the vast amount of literature around the scale (e.g., see introduction in Gnambs & Staufenbiel, 2018, Health Psychology Review, 12, 179-194), I am sure that the author can find terminology that avoids overlap with clinical diagnostic categories.

Response: Apologies for the confusion here – I tried not label GHQ-12 as ‘anxiety/depression’, for precisely the reasons that you give! There are one or two places in the original text where I inadvertently got close to doing this however, and I have corrected these.

There is however a second mental health measure available in HSE which I label as ‘anxiety/depression’, which I describe in the web appendices as follows:

In the self-completion survey in 1996, 2003-6, 2008, 2010-12 and 2014, respondents were asked ‘Now we would like to know how your health is today. Please answer ALL the questions. By ticking one box for each question below, please indicate which statements best describe your own health state today’:

- “I am not anxious or depressed”
- “I am moderately anxious or depressed”
- “I am extremely anxious or depressed”

[This is part of the widely-used EQ-5D health status indicator 5. However, for the purposes of this paper we have separated the individual measures that make up the EQ-5D in order to compare these to similar indicators of morbidity within each domain].

Two outcome measures are based on this: whether people have any anxiety/depression (the 2nd and 3rd categories combined), and whether they have extreme anxiety/depression (3rd category only).

In the light of your comments, I have changed the text to emphasise that this is people’s self-reports of feeling anxious or depressed, and not a clinical diagnosis.

Finally, the references on GHQ-12 you provide are helpful (I have a wider interest in CAT methods, so it’s interesting to see your use of it in this context). I much prefer your term ‘psychological distress’ to the term I previously used, ‘minor psychiatric morbidity’ – I have amended the text as a result and included a reference to your paper in the appendices.

R3: CRP and fibrinogen are not commonly used measures of heart disease risk (for that other biomarkers would be needed, e.g. Troponin). I would suggest to turn the presentation around and state that both are generic markers of inflammatory processes and have been linked to (and then the diseases and disorders with the references as presented in the paragraph).

Response: Many thanks for this suggestion – I have changed the text as you suggest.

Minor comments

R1: Throughout “incidence” is used when author is actually talking about prevalence.

R1: Presentation of the tables: ...Column heading “Incidence” when from methods and text in results it sounds like change in prevalence was assessed.

Response: many thanks for spotting this; this has now been corrected.

R3: A minor point, but related to this: I think in Appendix I a lot of references should actually point to Appendix III?

Response: Apologies for this – I have checked the main text and appendices to ensure that references to appendix numbers are correct. (I was required during the submission process to re-number the appendices in the order in which they appear in the text, at which point occasional numbering errors were introduced).

R3: The code should not be hosted on the author's web page, since these are subject to frequent changes. Since many different repositories exist (OSF, github,...) that are for exactly this purpose, the code can easily be shared over one of those pages. [I do not think this is an issue since the author already indicated that it will be shared]

Response: I have set up an OSF account as follows:

Geiger, B. B. (2019, August 26). Morbidity in England 1994-2014. Retrieved from <http://osf.io/dy6sv>

R3: Without sharing the code with the review it is really difficult to evaluate (a) whether all code and data have been shared and (b) how appropriate the applied methods were. [I do not think this is an issue since the author already indicated that it will be shared]. General and maybe helpful reminder for [this comment]: BMJ Open review is not blind, so sharing the code is possible through any repository (most would even allow an anonymous link anyway).

Response: I am torn here. On the one hand, I am committed to sharing and transparency – I make any data I collect publicly accessible, and I am committed to making my code publicly accessible for others to use (and to improve/correct), without restriction beyond attribution. In this case, I very much hope that the HSE time-series dataset I have created will be a valuable resource for other researchers.

On the other hand, I am not sure that my code should be reviewed as part of the journal submission process. Partly this is because it seems like an excessive burden to put reviewers through – my code is nearly 3000 lines long. But partly it seems that this is unusual even within open access journals such as BMJ Open, and there is the potential that the additional material subjected to review makes it harder for me to publish my work, compared to others who are less committed to transparency and open collaboration.

Given that I am torn, I have left it for the reviewer to decide how to proceed here: I have put my code up at the OSF link above, and if you feel that it is both possible and necessary to review the code, then you are able to do so.

R3: Author statement unclear: "A research assistant (Mariska van der Horst) briefly helped with some of the data preparation." The named researcher is not an author and would not qualify according to ICMJE criteria. The input is noted appropriately under "Acknowledgements".

Response: as suggested, I have taken this out of the author statement.

R3: I think generally the manuscript could do with a read through for adjectives and comparatives. They can largely be removed. It is a detailed descriptive paper. It is for the readers to assess whether an increase or decline of something is particularly "sharp" or "marked". Without clear guidelines on something is interpreted as such, it is subjective and there is no need to qualify the reported numbers further.

Response: this comment is well-taken – but with a caveat.

There is a well-known danger that patterns that are statistically significant are interpreted as practically significant – even when the size of effect is very small. Some (e.g. Andrew Gelman, who has written about 'Type M' errors about the magnitude of effects at <https://doi.org/10.1177%2F1745691614551642>) even argue that there is nothing that we look at where the true effect is zero; the question is whether the effect is large enough that we should care about it.

As such, I think it is important to make clear in the discussion whether a change over time is large or small – this is necessary to provide an adequate description of what is occurring

(even in the absence of any normative judgements). For example, the conclusions relating to Table 1 are about the different sizes of different trends.

That said, there is no need for me to use terms like 'sharp' or 'considerable' – so I have changed my use of these simply to 'large'. My feeling is that these are inevitably subjective terms (because the practical significance of an effect is a subjective judgement), but they are now hopefully not more value-laden than they need to be.

R3: page 6, "(which are unreliable, as we explain below)" This is a throwaway statement here and should be removed. The discussion further down in the manuscript should be enough.

Response: I have re-written this section of the manuscript in line with the comments from Reviewer 2 above.

R3: page 7: "which provide further insight into whether reported changes are simply reporting changes". This is unclear to me and could maybe rephrased?

Response: apologies for the confusing wording; I have rephrased this.

R3: Appendix 4: "For several of the general health measures, there is more evidence of change over this period..." 'More' compared to what?

Response: apologies for the confusing wording; I have rephrased this.

R3: Appendix 4: "As an aside, UK Government publications have made claims based on healthy/disability-free life expectancy, most recently to argue that morbidity has been deteriorating." Please provide references and point clearly to those claims. Otherwise this reads like opinion/conjecture.

Response: apologies for the lack of references; they have now been added.

R3: Page 22, "but deteriorating obesity, diabetes, some biomarkers (fibrinogen and possibly also CRP) and mental ill-health at the highest levels". Unclear what "we see deteriorating x at highest levels" means

Response: apologies for the confusing wording; I have rephrased this.

R3: Page 23, typo: symptom_a_tology?

Response: many thanks for spotting this typo.

Gender-stratified models for R1

Additional response: This table shows the difference between the male (vs. female) trends in each of the morbidity indicators used in the main paper. This is based on a similar logistic regression model to the main paper, except that it adds the interaction of gender and period, and to simplify the age-sex adjustment, only adjusts for age and not age-sex interactions:

$$y_i = \text{logit} [\beta_1 \text{period}_i + \beta_2 \text{male}_i + \beta_3 (\text{period}_i * \text{male}_i) + \beta_4 \text{age}_i]$$

...where period_i refers to a vector of period dummy variables, male_i refers to a binary gender dummy variable, and β_3 is a vector of the key outcome coefficients showing the difference between men and women in the change between each period and the earliest available period.

As in the main paper, the figures are given as average marginal effects. For example, the first result below shows that the change over time in 'high blood pressure LSI' for men is 1.2% (95% CI 0.3 to 2.2%) more positive than for women. This is summarised in the response to the reviewer above.

	Male vs. female change over time estimate (95% confidence interval)	
HEART & CIRCULATORY		
Blood pressure/cholesterol		
High blood pressure LSI	1.2%	(0.3%, 2.2%)
Recent high blood pressure	4.3%	(2.7%, 5.9%)
Biomarker high blood pressure	-4.6%	(-5.6%, -3.5%)
High total cholesterol	3.4%	(0.7%, 6.1%)
Low HDL cholesterol	-6.9%	(-8.6%, -5.2%)
Other CVD		
Recent heart attack/stroke	0.1%	(-0.5%, 0.7%)
Recent angina	-0.1%	(-0.6%, 0.5%)
IHD/stroke LSI	-0.3%	(-0.8%, 0.2%)
Heart attack symptoms	-0.6%	(-2.0%, 0.9%)
Mini stroke (TIA) symptoms	0.4%	(-1.3%, 2.1%)
Angina symptoms	0.1%	(-0.8%, 0.9%)
Any CVD LSI	0.9%	(-0.4%, 2.1%)
Any recent CVD	0.4%	(-0.8%, 1.6%)
RESPIRATORY		
COPD symptoms	0.3%	(-1.1%, 1.6%)
Diagnosed asthma	-0.2%	(-2.4%, 2.0%)
Asthma LSI	0.1%	(-1.4%, 1.5%)
Breathlessness-Grade 2+	1.8%	(-0.4%, 4.0%)
Breathlessness-Grade 3	0.5%	(-1.0%, 2.1%)
Recent wheezing/asthma	0.4%	(-2.0%, 2.8%)
Wheezing stopping sleep	-0.2%	(-1.2%, 0.9%)
ANTHROPOMETRIC & DIABETES		
BMI-Underweight	0.3%	(-0.1%, 0.7%)
BMI-Obese	1.7%	(0.1%, 3.2%)
	-	
High waist-hip ratio	13.1%	(-15.0%, -11.2%)
Recent diabetes	0.6%	(-0.1%, 1.2%)
Diabetes LSI	0.5%	(-0.4%, 1.5%)
Glycated haemoglobin	1.6%	(0.4%, 2.8%)

MENTAL HEALTH

Mental health LSI	-1.0%	(-2.2%, 0.2%)
Psychiatric morbidity symptoms	1.0%	(-0.9%, 2.8%)
Anxiety/depression-moderately	0.5%	(-1.6%, 2.5%)
Anxiety/depression-extremely	-0.2%	(-1.0%, 0.5%)
High psychiatric morbidity	0.5%	(-0.5%, 1.5%)

ACTIVITY LIMITATIONS & MUSCULOSKELETAL

Problems walking about today	-0.1%	(-1.7%, 1.4%)
Problems washing/dressing today	-0.2%	(-1.2%, 0.7%)
Any locomotor limitation	0.6%	(-0.8%, 2.0%)
Any self-care limitation	0.5%	(-0.6%, 1.6%)
Pain-any	0.3%	(-1.9%, 2.6%)
Pain-extreme	-0.5%	(-1.3%, 0.4%)
Arthritis LSI	1.2%	(-0.1%, 2.4%)
Other musculoskeletal LSI	-2.3%	(-4.0%, -0.6%)

OTHER BIOMARKERS

Raised C-reactive protein	-0.3%	(-5.3%, 4.7%)
Raised fibrinogen	0.2%	(-2.2%, 2.6%)
Anaemia	0.5%	(-2.3%, 3.3%)
Iron deficiency	7.8%	(3.1%, 12.5%)

VERSION 2 – REVIEW

REVIEWER	Sarah Cook UiT, The Arctic University of Norway, Norway
REVIEW RETURNED	04-Nov-2019

GENERAL COMMENTS	No further comments - all previous comment have been thoroughly addressed.
--

REVIEWER	Coste, Joel Biostatistics and Epidemiology Unit, Hôtel Dieu, Assistance Publique-Hôpitaux de Paris
REVIEW RETURNED	05-Oct-2019

GENERAL COMMENTS	The paper has greatly improved. However, it still lacks explicit research questions and conceptual framework. Morbidity is not a simple concept but a difficult, still debated and probably multidimensional one. The Introduction should provide the theoretical background of the study. By the way, how can the author state he "does not have formal hypotheses" and presents "a descriptive paper" of "changes over time" when the first part of its title is "Has working-age morbidity been declining?"?
---

REVIEWER	Jan R Boehnke University of Dundee
REVIEW RETURNED	24-Oct-2019

GENERAL COMMENTS

The authors of "Has working-age morbidity been declining? Trends in general health, chronic diseases, symptoms and biomarkers in England 1994-2014" (bmjopen-2019-032378) Have made multiple and detailed adjustments in response to the reviewers' comments. Especially changes made regarding the previous comments relating to trends vs. changes are well-presented. Readability and clarity have improved and as stated in the first round, it remains an immensely interesting paper.

All page numbers are the ones provided by the authors in their version with track changes.

The code was not accessible as requested or promised by the manuscript. It may be deposited on OSF, but it is password protected, i.e. one has to log on to OSF. This was therefore still not evaluated, but the presentation of the statistical analysis was greatly helped by the additions made.

__ MAJOR comments __

1) Additions were made on page 8 of which I am not sure why and which seem not helpful. Particularly this refers to:
„...suffer from what is variably conceptualised as ‘response shift’²⁸ or ‘differential item functioning’²⁹; that is, for any given question, different people (or even the same people at different times) report their general health/disability on different scales.“

a) Response shift and differential item functioning are not the same thing and the current sentence implies this.

b) Response shift is an intra-individual phenomenon, especially as presented in the referenced literature of this manuscript. It can therefore not be relevant for this analysis that only deals with inter-individual differences.

c) Differential item functioning, largely speaking, cannot be investigated for single items, i.e. it is not relevant at this position in the paper that deals with single indicators.

d) Even if this were true, neither (DIF, response shift) is an indication that people report their health on different scales (unless the questionnaire would suddenly change its response format).

e) As stated, I do not understand why this was inserted (reviewer commentaries and responses by the authors were not shared with reviewers), i.e. I would just describe that multiple indicators are available and that they will descriptively be compared (which seems 100% in line with the research question, title of the manuscript etc pp). It reads like the attempt of a justification to state why also other measures are used as indicators of health, but (a) as an epidemiologist I feel that this is not necessary since the relevance of evaluating multiple indicators seems evident and (b) as a psychometrician and response shift researcher I feel that this attempt is based on wrong assumptions and does not reflect what the terms used are actually describing in their respective research strands.

2) Page 8, „These inconsistent response scales mean that general health/disability measures are inadequate for answering our question“

As described in the previous comment, I think the previous passage in the manuscript is wrong, therefore also the conclusion from that passage presented in this sentence is wrong (or does not follow). If this is what the authors want to say, then using classic survey literature on the cognitive interpretation of questions (e.g., work by Torangeau and colleagues) and the construction of the fact that the interpretation of questions differs across historic times and (sub)cultures should suffice to justify (at least in the eyes of the authors) the downplaying of the evidence derived from such general questions.

3) page 11, „they seem substantially less likely to be affected by changing medical practice than label-based measures, and are therefore likely to be more comparable over time.“

After the downplaying of the usefulness of other general health items, this seems a bit slapdash and shorthand. Why exactly are these items better and where is the literature backing this up?

4) In line with the authors argument, page 11 „biomarkers are most reliable in measuring changes over time, but do not capture morbidity well“ could maybe read

„biomarkers are least dependent on observer and/or respondent interpretation over time, but do not capture morbidity well

5) Page 26: “but there is no sign that this has occurred; for example, trends in education are similar in HSE and the gold-standard measure of qualification trends, the Labour Force Survey.”

This observation seems equally plausible if both of these surveys were subject to (changing trends in) non-response bias, which seems quite likely. I would just scrap this argument, since without controlling for education (and other potential confounders in sampling design) in the regression models, via sample-weighting or with a selection model, this cannot be addressed. The limitation should just left standing without qualification as it is just the nature of such survey/secondary analysis work.

__ MINOR comments __

Page 11: „Rose angina scale, GHQ psychiatric distress“
References should be provided.

Table 3: Maybe add a note to “Psychological Distress” and refer to GHQ12 in note to table?

All tables: “LSI” needs to be de-abbreviated in the note to table.

Page 12: “Table 3 also confirms a large rise in diabetes”
According to the table these are 2.1%. This does not seem to be a “large” rise.

More generally the manuscript could be re-read regarding the use of such adjectives. In scientific writing they are generally not needed since it is for the readers to make judgements about the size/ importance of the reported results – or for the authors to

	make a case why the use of an adjective such as large/sharp is appropriate. Page 21: "Adult Psychiatric Morbidity Surveys" Citation missing.
--	--

VERSION 2 – AUTHOR RESPONSE

Response to reviewers

Reviewer 1: Sarah Cook

Reviewer 2: Joel Coste

Reviewer 3: Jan R. Boehnke

Author's note: as before, I have responded to all of your comments individually, but I have re-ordered the comments so that I respond to similar points at the same time (even if from different reviewers). This should hopefully make it easy to see how I have responded to your specific comments, at the same time as seeing how this fits with responses to other reviewers.

General comments

R1: No further comments - all previous comment have been thoroughly addressed.

R2: The paper has greatly improved.

R3: The authors of "Has working-age morbidity been declining? Trends in general health, chronic diseases, symptoms and biomarkers in England 1994-2014" (bmjopen-2019-032378) have made multiple and detailed adjustments in response to the reviewers' comments. Especially changes made regarding the previous comments relating to trends vs. changes are well-presented. Readability and clarity have improved and as stated in the first round, it remains an immensely interesting paper.

Author: I am glad that you feel that my previous responses have helped the paper! I appreciate the time you have spent as reviewers in helping me improve the paper, notwithstanding the changes that remain to be made below.

Response shift / DIF

R3: Additions were made on page 8 of which I am not sure why and which seem not helpful. Particularly this refers to: „...suffer from what is variably conceptualised as ‘response shift’²⁸ or ‘differential item functioning’²⁹; that is, for any given question, different people (or even the same people at different times) report their general health/disability on different scales.“

a) Response shift and differential item functioning are not the same thing and the current sentence implies this. b) Response shift is an intra-individual phenomenon, especially as presented in the referenced literature of this manuscript. It can therefore not be relevant for this analysis that only deals with inter-individual differences. c) Differential item functioning, largely speaking, cannot be investigated for single items, i.e. it is not relevant at this position in the paper that deals with single indicators. d) Even if this were true, neither (DIF, response shift) is an indication that people report their health on different scales (unless the questionnaire would suddenly change its response format).

e) As stated, I do not understand why this was inserted (reviewer commentaries and responses by the authors were not shared with reviewers), i.e. I would just describe that multiple indicators are

available and that they will descriptively be compared (which seems 100% in line with the research question, title of the manuscript etc pp). It reads like the attempt of a justification to state why also other measures are used as indicators of health, but (a) as an epidemiologist I feel that this is not necessary since the relevance of evaluating multiple indicators seems evident and (b) as a psychometrician and response shift researcher I feel that this attempt is based on wrong assumptions and does not reflect what the terms used are actually describing in their respective research strands.

Even if this were true, neither (DIF, response shift) is an indication that people report their health on different scales (unless the questionnaire would suddenly change its response format).

Author: many thanks for these detailed comments – I have tried to reply in similar detail, as I can see that this is an issue that you feel strongly about.

On response shift

I should start by noting that the discussion of response shift was done in response to another reviewer (R2), who said, “Discuss further why people may respond differently over time to identical questions and relate this problem to that often referred to in the literature as ‘response shift’.”

I understand the difference in perspective between you and R2 here. R2 was indicating that I should bring in a relevant literature, while you are arguing that my presentation of response shift is (while potentially relevant) somewhat misleading. You mention several aspects to this, each of which requires a different response:

- *My apologies for implying that response shift and DIF are the same thing – this was not my intention, but I take your comment that the sentence is currently misleading. I have changed this (see also below on DIF).*
- *One issue here is my use of the phrase “report their general health/disability on different scales.” I think you interpreted this to mean different explicit response categories – and as you say, this cannot happen unless the questionnaire suddenly changes its response format. However, I meant that respondents are relating the response categories differently to underlying measurement scales.*
- *Hopefully my comment now makes more sense – response shift CAN be thought of as a change in how individuals relate their health to the response categories in a given survey question. (For example, some respondents report their self-reported health compared to other people of their age (Adamson et al., 2004; Mallinson, 2002), which may well lead to response shift as they age).*
- *This fits the definition of response shift referred to in Sprangers et al 1999 cited in the paper, who define response shift as “a change in the meaning of one’s self-evaluation of a target construct as a result of: (a) a change in the respondent’s internal standards of measurement (scale recalibration, in psychometric terms); (b) a change in the respondent’s values (i.e. the importance of component domains constituting the target construct); or (c) a redefinition of the target construct (i.e. reconceptualization).” Sprangers et al later clarify this in the example of Ann, diagnosed with Stage 3 breast cancer, who “might engage in downward social comparison, by talking with and supporting less fortunate other people with a similar diagnosis. These behaviors might induce response shift due to changes in internal standards, since her idea of poor functioning may now be anchored at a much lower level than previously conceived.”*
- *I should stress: I completely take the point that there are other aspects of response shift (as the model of Sprangers et al makes clear), and response shift is not*

reducible to reporting changes with respect to a stable underlying concept. Nevertheless, I take R2's point that response shift is relevant to the point that I am making.

To try to balance these different positions, I have kept the reference to response shift, but tried to reword this to avoid any potential misunderstandings. Hopefully the revised wording manages to link my argument to the wider literature on response shift, without being misleading in the way that you draw attention to in your comments. Finally, it is worth noting that this is a small and non-essential part of my argument, so I very much hope it is possible to develop a form of wording that satisfies the valid comments of both yourself and Reviewer 2.

On DIF

Reviewer 2 did not ask me to include a reference to DIF, and given your concerns, I have simply removed any reference to DIF in the paper. (I wrote an explanation for why I had originally included this, but on reflection I don't think you need to read this, given that I have taken this out!).

R3: 2) Page 8, „These inconsistent response scales mean that general health/disability measures are inadequate for answering our question“ As described in the previous comment, I think the previous passage in the manuscript is wrong, therefore also the conclusion from that passage presented in this sentence is wrong (or does not follow). If this is what the authors want to say, then using classic survey literature on the cognitive interpretation of questions (e.g., work by Torangeau and colleagues) and the construction of the fact that the interpretation of questions differs across historic times and (sub)cultures should suffice to justify (at least in the eyes of the authors) the downplaying of the evidence derived from such general questions.

Author: hopefully the previous response clarifies how this argument DOES follow: general health/disability questions do not clearly specify the scale that respondents report against. (Put more concretely: different people (or even the same person at different times) will draw the line between 'good' and 'very good' health differently). As a result, general health/disability measures are not a good basis for investigating morbidity trends over time – and this is because of the reporting as well as the comprehension element of Tourangeau et al's model of the survey response process.

However, I very much take the point that the discussion was not worded well. Hopefully you are happier with the revised wording here, which as suggested includes a reference to Tourangeau et al's cognitive model of the survey response process (via the comprehensive Groves et al review of such models), as well as the revised wording of the response scale issues.

R3: 3) page 11, „they seem substantially less likely to be affected by changing medical practice than label-based measures, and are therefore likely to be more comparable over time.“ After the downplaying of the usefulness of other general health items, this seems a bit slapdash and shorthand. Why exactly are these items better and where is the literature backing this up?

Author: this point is well-taken – I had made this argument in another point in the text, and the structure of the argument here was not well-constructed. I have changed the structure of this section so that the justification for the superiority of symptom-based measures is made at this point (and furthermore draws attention to the limitations that remain).

R3: 4) In line with the authors argument, page 11 „biomarkers are most reliable in measuring changes over time, but do not capture morbidity well“ could maybe read „biomarkers are least dependent on observer and/or respondent interpretation over time, but do not capture morbidity well

Author: I have now changed this to something very similar to the wording suggested by the reviewer (to: biomarkers are “least likely to be affected by changing respondent interpretations over time, but do not capture morbidity well”).

Research questions

R2: However, it still lacks explicit research questions and conceptual framework. Morbidity is not a simple concept but a difficult, still debated and probably multidimensional one. The Introduction should provide the theoretical background of the study.¹

Author: as I said before, I do see this as primarily a descriptive paper. Indeed, the closest recent paper by Reviewer 2 that I can find (<http://dx.doi.org/10.1136/jech-2018-210941>) – which is enormously useful and now cited in the paper – similarly contains no explicit hypotheses. Having read many such papers, it is simply not conventional to have explicit research questions / hypotheses.²

Nevertheless, I know that the Reviewer feels strongly about this, and I have tried to respond to this comment as well as I can by:

- *Explicitly talking about three competing hypothesis (of morbidity decline, morbidity stability or morbidity improvement) in the introduction of the paper and in the conclusions.*
- *Explicitly talking about the multidimensional nature of morbidity in the Introduction, the Measures section, and the Conclusion – linking my use of multiple specific morbidity indicators to the multidimensional nature of morbidity, but also how these can be used to answer the broad hypotheses about morbidity trends in general.*

Space has precluded me from long text additions (indeed, it was very hard to stay within the word limit while responding to the two rounds of comments) – but hopefully the new wording is an improvement, and sufficient to provide some reassurance around your concerns here.

R2: By the way, how can the author state he "does not have formal hypotheses" and presents "a descriptive paper" of "changes over time" when the first part of its title is "Has working-age morbidity been declining?" ?

Author: apologies for the confusion here – the title refers to the hypothesis (/assumption) of policymakers that morbidity has been declining; my paper challenges this assumption and seeks to describe what has actually happened. For this reason I see it as a descriptive paper, but as I said in response to the previous comment, I am happy set out the competing hypotheses (and my a priori ambivalence) more explicitly in the paper.

Response biases

R3: 5) Page 26: “but there is no sign that this has occurred; for example, trends in education are similar in HSE and the gold-standard measure of qualification trends, the Labour Force Survey.” This observation seems equally plausible if both of these surveys were subject to (changing trends in) non-response bias, which seems quite likely. I would just scrap this argument, since without controlling for

¹ Note the previous comment: R2: *The manuscript is generally well written but the authors should more clearly state 1) the objectives of the study and hypotheses; 2) how the methods and results align with those objectives; 3) how the different measures may provide converging, diverging or complementary evidence. The respective place of the various measures of global/focal health/disability should be better and earlier defined.*

² As I said before, I also do not perform conventional statistical significance testing for reasons given in the paper itself, and deliberately avoid binary rejection/acceptance of the null hypothesis. There are many defensible responses to the current challenges in significance testing, but hopefully you agree that this is one of several reasonable responses.

education (and other potential confounders in sampling design) in the regression models, via sample-weighting or with a selection model, this cannot be addressed. The limitation should just left standing without qualification as it is just the nature of such survey/secondary analysis work.

Author: this point is well-taken; this sentence has been removed.

Code

R3: The code was not accessible as requested or promised by the manuscript. It may be deposited on OSF, but it is password protected, i.e. one has to log on to OSF. This was therefore still not evaluated, but the presentation of the statistical analysis was greatly helped by the additions made.

Author: my sincere apologies (this was my first time using OSF...). I have now made the OSF code public.

Minor comments

R3: Page 11: „Rose angina scale, GHQ psychiatric distress“ References should be provided.

Author: These have been added.

R3: Table 3: Maybe add a note to “Psychological Distress” and refer to GHQ12 in note to table?

Author: This has been added.

R3: All tables: “LSI” needs to be de-abbreviated in the note to table.

Author: This has been added.

R3: Page 12: “Table 3 also confirms a large rise in diabetes”. According to the table these are 2.1%. This does not seem to be a “large” rise. More generally the manuscript could be re-read regarding the use of such adjectives. In scientific writing they are generally not needed since it is for the readers to make judgements about the size/ importance of the reported results – or for the authors to make a case why the use of an adjective such as large/sharp is appropriate.

Author: The rise in diabetes as measured via a biomarker (glycated haemoglobin) is 2.1% (percentage points) from 2001-3 to 2011-14, against a baseline rate of 2.7%. My view is that a near-doubling of diabetes does therefore count as a ‘large rise’.

I take your previous point that excessive use of adjectives (e.g. ‘marked’ or ‘sharp’) is not appropriate in a research paper. However, as I said previously (though I am not sure if you saw my previous response, given your comment above), there is a well-known danger that patterns that are statistically significant are interpreted as practically significant – even when the size of effect is very small. Some (e.g. Andrew Gelman, who has written about ‘Type M’ errors about the magnitude of effects at <https://doi.org/10.1177%2F1745691614551642>) even argue that there is nothing that we look at where the true effect is zero. Rather, the question that researchers face is whether the effect is large enough that we should care about it.

As such, I think it is important to make clear in the discussion whether a change over time is large or small – this is necessary to provide an adequate description of what is occurring (even in the absence of any normative judgements). For example, the conclusions relating to Table 1 are about the different sizes of different trends. I have changed all of the terms to ‘large’ (rather than e.g. ‘considerable’); my feeling is that these are inevitably subjective terms (because the practical significance of an effect is a subjective judgement), but they are now hopefully not more value-laden than they need to be.

R3: Page 21: “Adult Psychiatric Morbidity Surveys”: Citation missing.

Author: This has been added.

Bibliography for 2nd response to reviewers

- Adamson, J., Gooberman-Hill, R., Woolhead, G., and Donovan, J. (2004). 'Questerviews': using questionnaires in qualitative interviews as a method of integrating qualitative and quantitative health services research *Journal of Health Services Research and Policy*, 9(3), 139-145
- Mallinson, S. (2002). Listening to respondents: a qualitative assessment of the Short-Form 36 Health Status Questionnaire *Social Science & Medicine*, 54(1), 11-21

VERSION 3 – REVIEW

REVIEWER	Coste, Joel Biostatistics and Epidemiology Unit, Hôtel Dieu, Assistance Publique-Hôpitaux de Paris
REVIEW RETURNED	17-Dec-2019

GENERAL COMMENTS	The author has addressed my concerns.
---------------------------------------

REVIEWER	Jan R. Boehnke University of Dundee, UK
REVIEW RETURNED	06-Dec-2019

GENERAL COMMENTS	It was interesting to look at this already very good manuscript again. 1) The code was made available now and although I am a fan of depositing code for analyses as an appendix with the journal, since this would not allow authors to remove the code at some later point/ make it unavailable again, in this case it seems maybe too much for a long appendix. On the other hand BMJOpen is an online journal and could potentially afford an online appendix. I would follow editor's advice on this since I am not familiar with the journal's policy in this matter. I was not able to run the analyses because this would take several hours of preparation alone, but I looked through the appendices and the analyses seem all documented to an excellent amount of detail, with good explanations, visuals and in a logical manner. There are some typos in the descriptive commentary of the documents, but given the length of the documents I applaud the author for creating an excellent supporting resource. I rely on the assertion that the code was fully re-run by the author on the 26th August 2019 and the fact that this is so much more than is usually provided by authors. There was one issue with the following document, 0_HSE_master_public.do, which did not open. 1) The passage around the use of single indicator measures has been comprehensively revised (pages 8/9 of the document). Nevertheless, one issue remains. The text uses the word "unidimensional", which is incorrect. The author discusses here _single indicator_ or _single question_ assessments. But just because you have one variable, does not mean that the responses to this variable are unidimensional – which is in general a statement about the number of causally contributing factors that determine the response to that variable. Especially questionnaire data, multidimensionality cannot occur only between items (i.e.
---

	leading to multiple factors in multi-item assessment instruments), but also within one item (i.e. multiple meaningful dimensions contributing to/determining the response distribution; e.g., https://journals.sagepub.com/doi/full/10.1177/0013164415575147). The author acknowledges this point explicitly in this passage as well (“Numerous causal factors contribute to this variable comprehension/reporting, ranging...”). I would therefore suggest the following changes: (i) ...are a simple way of measuring morbidity __with a single indicator__. (instead of using unidimensionally) (ii) Change the end of the last paragraph from “Moreover, unidimensional measures are potentially misleading in that they gloss over the multidimensional nature of morbidity.1” to say rather “Moreover, __single indicator__ measures are potentially misleading in that they gloss over the multidimensional nature of morbidity.1” [or any other change to that effect]
--	---

VERSION 3 – AUTHOR RESPONSE

Response to reviewers

Reviewer 1: Sarah Cook

Reviewer 2: Joel Coste

Reviewer 3: Jan R. Boehnke

Author’s note: as before, I have responded to all of your comments individually, but I have re-ordered the comments so that I respond to similar points at the same time (even if from different reviewers).

General comments

R2: The author has addressed my concerns.

Author: I am glad that the previous revisions satisfied your concerns – thanks once more for reviewing the paper!

R3: It was interesting to look at this already very good manuscript again.

Author: Many thanks to for looking at the paper in detail once more!

Supplementary files

R3: 1) The code was made available now and although I am a fan of depositing code for analyses as an appendix with the journal, since this would not allow authors to remove the code at some later point/ make it unavailable again, in this case it seems maybe too much for a long appendix. On the other hand BMJOpen is an online journal and could potentially afford an online appendix. I would follow editor’s advice on this since I am not familiar with the journal’s policy in this matter. I was not able to run the analyses because this would take several hours of preparation alone, but I looked through the appendices and the analyses seem all documented to an excellent amount of detail, with good explanations, visuals and in a logical manner. There are some typos in the descriptive commentary of the documents, but given the length of the documents I applauded the author for creating an excellent supporting resource. I rely on the assertion that the code was fully re-run by the

author on the 26th August 2019 and the fact that this is so much more than is usually provided by authors.

Editorial Requests: We note that reviewer 3 has suggested that you upload your raw code as supplementary files to BMJ Open. We would advise that they are kept at the external address provided as they are very extensive.

Author: following the editorial request, I have kept the raw code at the external address.

R3: There was one issue with the following document, 0_HSE_master_public.do, which did not open.

Author: apologies for this. I have fixed the issue with OSF simply by changing the filename (having 'public' in a file name seems to lead the file to corrupt).

'Unidimensional'

R3: 1) The passage around the use of single indicator measures has been comprehensively revised (pages 8/9 of the document). Nevertheless, one issue remains. The text uses the word "unidimensional", which is incorrect. The author discusses here *_single indicator_ or _single question_ assessments*. But just because you have one variable, does not mean that the responses to this variable are unidimensional – which is in general a statement about the number of causally contributing factors that determine the response to that variable. Especially questionnaire data, multidimensionality cannot occur only between items (i.e. leading to multiple factors in multi-item assessment instruments), but also within one item (i.e. multiple meaningful dimensions contributing to/determining the response distribution; e.g., <https://journals.sagepub.com/doi/full/10.1177/0013164415575147>). The author acknowledges this point explicitly in this passage as well ("Numerous causal factors contribute to this variable comprehension/reporting, ranging..."). I would therefore suggest the following changes:

(i) ...are a simple way of measuring morbidity *__with a single indicator__*. (instead of using unidimensionally)

(ii) Change the end of the last paragraph from "Moreover, unidimensional measures are potentially misleading in that they gloss over the multidimensional nature of morbidity.1" to say rather "Moreover, *__single indicator__* measures are potentially misleading in that they gloss over the multidimensional nature of morbidity.1"

[or any other change to that effect]

Author: many thanks for this comment – I completely take your point, and have amended the text as suggested.

Title

Editorial Requests: Please revise the title to indicate the research question, setting, and study design. This is the preferred format for the journal.

Author: I have revised the title to indicate that the paper uses survey measures. It now includes the research question ('Has working-age morbidity been declining?'), study design ('Changes over time in survey measures of general health, chronic diseases, symptoms and biomarkers'), and setting ('in England 1994-2014').